# Optimal and Provable Calibration in High-Dimensional Binary Classification: Angular Calibration and Platt Scaling

**Yufan Li**
Department of Statistics
Harvard University
Cambrdige, MA 02138
yufan_li@g.harvard.edu

**Pragya Sur**
Department of Statistics
Harvard University
Cambrdige, MA 02138
pragya@fas.harvard.edu

## Abstract

We study the fundamental problem of calibrating a linear binary classifier of the form $\sigma(\hat{w}^\top x)$, where the feature vector $x$ is Gaussian, $\sigma$ is a link function, and $\hat{w}$ is an estimator of the true linear weight $w^\star$. By interpolating with a noninformative *chance classifier*, we construct a well-calibrated predictor whose interpolation weight depends on the angle $\angle(\hat{w}, w_\star)$ between the estimator $\hat{w}$ and the true linear weight $w_\star$. We establish that this angular calibration approach is provably well-calibrated in a high-dimensional regime where the number of samples and features both diverge, at a comparable rate. The angle $\angle(\hat{w}, w_\star)$ can be consistently estimated. Furthermore, the resulting predictor is uniquely *Bregman-optimal*, minimizing the Bregman divergence to the true label distribution within a suitable class of calibrated predictors. Our work is the first to provide a calibration strategy that satisfies both calibration and optimality properties provably in high dimensions. Additionally, we identify conditions under which a classical Platt-scaling predictor converges to our Bregman-optimal calibrated solution. Thus, Platt-scaling also inherits these desirable properties provably in high dimensions.

## 1 Introduction

Calibration of predictive models is a fundamental problem in statistics and machine learning, especially in applications that require reliable uncertainty quantification. A well-calibrated model ensures that its predicted probabilities align closely with true event probabilities—a property essential in fields such as medical decision-making [7, 41], meteorological forecasting [14, 28, 23, 65, 64], self-driving systems [60], and natural language processing [66, 29].

Numerous algorithms have been proposed for calibrating the outputs of a trained model, including classical methods such as Platt scaling [70, 13, 69, 31], histogram binning [89, 77], isotonic regression [90, 40, 12, 36, 44], and more recent approaches such as temperature scaling [29, 47], ensemble-based methods [50, 58, 86, 81], and Bayesian strategies [45, 21], among others.

While extensive prior work has studied calibration [48, 77, 30, 74, 43], this literature primarily focuses on traditional asymptotic theories or finite-sample learning theoretic arguments. These approaches often overlook the impact of problem dimensionality, which is particularly relevant for high-dimensional settings where the number of features may be substantial. Alternatively, a separate line of research has explored calibration within a high-dimensional proportional asymptotic regime, where the sample size $n$ and the feature dimension $d$ both diverge, at a comparable rate. This proportional scaling regime has gained significant traction in modern statistics and machine learning. In statistics, its popularity stems from the fact that theories derived under this regime

39th Conference on Neural Information Processing Systems (NeurIPS 2025).

capture high-dimensional phenomena observed in moderate to large sized datasets unusually well [42, 6, 25, 80, 91, 5, 78, 39, 63, 84, 52, 53]. Consequently, this has spurred the creation of innovative methods displaying remarkable practical performance [61, 27, 9, 54, 76, 57]. In machine learning, this regime has proven exceptionally valuable and effective in analyzing the behavior of modern neural networks and other interpolation learners under overparametrization [55, 34, 56, 59, 2, 75, 67]. For binary classification in this proportional regime, a substantial line of work [79, 78, 92] establishes that classical logistic regression yields seriously biased estimates; building upon these, [3] shows that logistic regression tends to be inherently overconfident, while [22] discusses the impact of regularization under the same model. Finally, [21] introduces expectation consistency and derives a limiting calibration error formula as a function of the signal prior and other problem parameters.

Despite these advancements, an approach that is provably calibrated in high dimensions, without knowledge of the true signal prior, is missing. Moreover, there is a lack of principled understanding regarding optimal calibration strategies from among the available options. Additionally, rigorous guarantees on the performance of classical calibration methods, such as Platt scaling, in modern high-dimensional scenarios is notably absent from the literature. In this paper, we address these gaps. We consider the challenge of calibrating a binary linear predictor in a frequentist setting under a Gaussian design. Our contributions are three-fold: (i) we introduce a data-driven predictor that can provably calibrate in a broad class of high-dimensional binary classification problems; (ii) we show that our calibrated predictor is *Bregman-optimal*, meaning it uniquely minimizes any Bregman divergence relative to the true label-generation probability; (iii) we establish conditions under which a classical Platt-scaled predictor converges to this Bregman-optimal calibrated solution, thereby formally showing that Platt scaling is both well-calibrated and Bregman optimal in our high-dimensional setting. Although we derive our theoretical results assuming Gaussian features, extensive recent universality results suggest that these should continue to hold for sufficiently light tailed distributions (see Section 8 for a discussion). We provide experiments that demonstrate this robustness to the Gaussian assumption (Section H.2 and 4).

We construct our calibrated predictor by interpolating with an uninformative ("chance") predictor, where the interpolation weight is determined by the angle $\angle(\hat{w}, w_\star)$ between the estimated linear weight $\hat{w}$ and the true weight $w_\star$. Our construction crucially leverages recent developments from the literature on observable estimation of unknown parameters in high dimensions. For instance, leveraging advances in [38, 8, 9, 10, 18, 54], we can show that the angle $\angle(\hat{w}, w_\star)$ is consistently estimable when $n$ and $d$ grow proportionally. To our knowledge, this is the first provable calibration method in a high-dimensional setting, and it uncovers a conceptual link between optimal calibration and $\angle(\hat{w}, w_\star)$: the poorer the alignment of $\hat{w}$ with $w_\star$, the greater the noise needed to be injected to prediction logits to ensure calibration.

## 2   Setting

Suppose we observe i.i.d. data $(y_i, x_i)$ satisfying

$$y_i \overset{\text{iid}}{\sim} \text{Bern}\left(\sigma\left(w_\star^\top x_i\right)\right), \quad i = 1, \ldots, n, \tag{1}$$

where $\sigma : \mathbb{R} \to [0, 1]$ denotes the link function and the covariates $x_i \in \mathbb{R}^d$ are drawn independently as $x_i \overset{\text{iid}}{\sim} N(0, \Sigma)$, with $\Sigma$ assumed to be known (say from a separate unlabeled dataset as in [17, 18]). The true linear weight $w_\star \in \mathbb{R}^d$ is an arbitrary deterministic vector, and we assume without loss of generality that $w_\star^\top \Sigma w_\star = \|w_\star\|_\Sigma^2 = 1$. The training dataset is denoted as $X = [x_1, \ldots, x_n]^\top \in \mathbb{R}^{n \times d}$ and $y = [y_1, \ldots, y_n]^\top \in \mathbb{R}^n$.

To quantify the degree of miscalibration, we define the calibration error at level $p$ for any predictor $\hat{f}$ as

$$\Delta_p^{\text{cal}}(\hat{f}) = p - \mathbb{E}_{x_{\text{new}}}\left[\sigma\left(w_\star^\top x_{\text{new}}\right) \mid \hat{f}\left(x_{\text{new}}\right) = p\right],$$

where $\mathbb{E}_{x_{\text{new}}}$ denotes the expectation over $x_{\text{new}} \sim N(0, \Sigma)$. A predictor is said to be *well-calibrated* if $\Delta_p^{\text{cal}}(\hat{f}) = 0$ for all $p$ in the range of $\hat{f}$. Intuitively, this means that when the predictor assigns a probability $p$ to label 1, the true probability of label 1 is indeed $p$.

We consider the regularized M-estimator

$$\widehat{w} = \arg\min_{w} \frac{1}{n} \sum_{i=1}^{n} \ell_{y_i} \left( w^{\top} x_i \right) + g(w),$$

where $g(\cdot)$ is a convex penalty and $\ell(\cdot)$ is a convex loss function. In this setting, we consider a sequence of problem instances $\{y(d), X(d), w_{\star}(d)\}_{d \geq 1}$ such that $X(d) \in \mathbb{R}^{n(d) \times d}$ and $y(d) \in \mathbb{R}^{n(d)}$ generated from (1). It is well-known that in the special case where $\ell(\cdot)$ equals the logistic loss and $g(\cdot)$ is zero, the corresponding predictor $\sigma(\widehat{w}^{\top} x_{\text{new}})$ is grossly mis-calibrated in the high-dimensional regime $\frac{n}{d} \to (0, +\infty)$ [79, 78, 3, 22], even where it is well-defined and unique [16].

In what follows, we present, for the first time, a predictor that is provably well-calibrated in this regime (for general convex losses and penalties beyond the special case mentioned above). We achieve this through an angular calibration idea, and furthermore, establish that this is optimal in the sense that it minimizes any Bregman divergence to the true label distribution. We conclude showing an interesting connection—Platt scaling converges to our angular predictor—and therefore is both provably well-calibrated and optimal in the aforementioned sense.

## 3    Introducing angular calibration

Most calibration strategies *adjust* a pre-trained predictor by learning a mapping $F \colon u \mapsto F(u)$ of the logits $\widehat{w}^{\top} x_{\text{new}}$. Platt scaling, for example, stipulates the parametric form $F(u) = \sigma(Au + B)$, where $A, B \in \mathbb{R}$ are fit on a holdout dataset. This raises a natural question:

*Among all well-calibrated predictors of the form $F(\widehat{w}^{\top} x_{\text{new}})$, which one is "the best"?*

In this section, we introduce a predictor with such an optimality property by interpolating between the prediction logits $\widehat{w}^{\top} x$ and an uninformative chance predictor. Specifically, we show that if the interpolation weight is determined by the angle between the estimator $\widehat{w}$ and the true weight $w_{\star}$, given by

$$\theta_* = \arccos \left( \frac{\langle w_{\star}, \widehat{w} \rangle_{\Sigma}}{\|\widehat{w}\|_{\Sigma} \|w_{\star}\|_{\Sigma}} \right), \tag{2}$$

then the resulting interpolated predictor minimizes any Bregman divergence to the true label distribution among all predictors of the form $F(\widehat{w}^{\top} x_{\text{new}})$. To the best of our knowledge, a predictor that is both provably calibrated and optimal (in the aforementioned sense) has not been previously introduced for high-dimensional problems.

Notably, our predictor uses the angle defined in (2), thus to define a data-driven predictor, we require a consistent estimate of this angle. Fortunately, recent advances in the high-dimensional literature (c.f., [38, 8, 9, 10, 18, 54]) allow us to estimate the inner product $\langle \widehat{w}, w_{\star} \rangle_{\Sigma}$, and therefore $\theta_*$, when $n$ and $d$ grow proportionally. We discuss the details of this estimation scheme later in Section 6. For now, we present our angular calibration idea assuming that we have access to a consistent estimator $\hat{\theta}$ for $\theta_*$.

**Definition 3.1.** (Angular Predictor) Let

$$\hat{f}_{\text{ang}} \left( \widehat{w}^{\top} x_{\text{new}}; \hat{\theta} \right) = \mathbb{E}_Z \left( \sigma \left( \cos\left(\hat{\theta}\right) \cdot \left( \frac{\widehat{w}^{\top} x_{\text{new}}}{\|\widehat{w}\|_{\Sigma}} \right) + \sin\left(\hat{\theta}\right) \cdot Z \right) \right) \tag{3}$$

where $\hat{\theta}$ is a consistent estimator of $\theta_*$ defined as in (13) and $\mathbb{E}_Z$ denotes expectation with respect to the *Gaussian noise* $Z \sim N(0, 1)$. We will later refer to $\hat{f}_{\text{ang}}$ as the angular predictor for simplicity.

Theorem 3.2 below shows that the angular predictor is well-calibrated. We defer the proof to Section A

**Theorem 3.2.** *Assume the link function $\sigma$ is continuous. Then, the predictor $\hat{f}_{\text{ang}}$ defined in (3) is well-calibrated as $d, n \to \infty, n/d \to (0, \infty)$. That is, for any $p$ contained in the range of $\sigma$, we have that*

$$\Delta_p^{\text{cal}} \left( \hat{f}_{\text{ang}} \left( \cdot; \hat{\theta} \right) \right) = p - \mathbb{E}_{x_{\text{new}}} \left[ \sigma \left( w_{\star}^{\top} x_{\text{new}} \right) \mid \hat{f}_{\text{ang}} \left( \widehat{w}^{\top} x_{\text{new}}; \hat{\theta} \right) = p \right] \to 0,$$

*in probability where $\hat{\theta}$ is a consistent estimator for $\theta_{\star}$ (Cf. Proposition 6.2).*

The above utilizes the result that when $\hat{\theta} = \theta_*$ exactly, $\hat{f}_{\text{ang}}(\cdot; \theta_*)$ is exactly well-calibrated (we state and prove this formally in Theorem A.1) and that $\hat{\theta}$ is consistent for $\theta_\star$. We will later show that $\hat{f}_{\text{ang}}(\cdot; \hat{\theta}) \approx \hat{f}_{\text{ang}}(\cdot; \theta_*)$ is in fact optimal in the sense that it minimizes any Bregman divergence to the true label distribution. The construction (3) admits an intuitive interpretation. By the basic trigonometric identity $\cos^2(\theta_*) + \sin^2(\theta_*) = 1$, we can see that the logits, that is, the argument of $\sigma(\cdot)$ in (3), is an interpolation between the informative component $\widehat{w}^\top x_{\text{new}}$ and the noninformative Gaussian noise $Z$. Notice that when $\widehat{w}$ is well aligned with $w_\star$ (i.e., $\cos^2(\theta_*) = 1$), the angular predictor $\hat{f}_{\text{ang}}$ lies closer to the informative predictor $\sigma(\widehat{w}^\top x_{\text{new}})$. Conversely, when $w_\star$ and $\widehat{w}$ are orthogonal (i.e., $\sin^2(\theta_*) = 1$), $\hat{f}_{\text{ang}}$ defaults to the non-informative chance predictor $\mathbb{E}[\sigma(Z)] = \mathbb{E}[\sigma(w_\star^\top x_{\text{new}})]$. In other words,

*The poorer the alignment between $w_\star$ and $\widehat{w}$, the greater the magnitude of noise $Z$ required to maintain calibration.*

We will show in the next section that this angular interpolation idea leads to a uniquely Bregman optimal calibrated predictor. This provides the first calibration procedure that is calibrated and optimal in high dimensions, provably. Peusdocode for angular calibration, using angle estimator from Section 6, is included in Section G.

# 4  Main Result I: Calibrating Optimally using Angular Calibration

Before formally stating our results on optimality of angular calibration, we first define the following random probability vectors for label distribution,

$$q_\star := \begin{pmatrix} \sigma(w_\star^\top x_{\text{new}}) \\ 1 - \sigma(w_\star^\top x_{\text{new}}) \end{pmatrix}, \quad \hat{q}_F := \begin{pmatrix} F(\widehat{w}^\top x_{\text{new}}) \\ 1 - F(\widehat{w}^\top x_{\text{new}}) \end{pmatrix}, \quad \hat{q}_{\text{ang}}(\hat{\theta}) := \begin{pmatrix} \hat{f}_{\text{ang}}(\widehat{w}^\top x_{\text{new}}; \hat{\theta}) \\ 1 - \hat{f}_{\text{ang}}(\widehat{w}^\top x_{\text{new}}; \hat{\theta}) \end{pmatrix}$$

(4)

where $F : \mathbb{R} \to [0, 1]$ is any measurable function. Here, $q_\star$ corresponds to the ground-truth probability distribution of the new label, $\hat{q}_F$ the prediction probability distribution of an $F$-calibrated predictor, and $q_{\text{ang}}$ the prediction probability distribution of our angular predictor.

Next, we define the Bregman loss function.

**Definition 4.1** (Bregman Loss Functions). Let $\phi : \mathbb{R}^2 \mapsto \mathbb{R}$ be a strictly convex differentiable function. Then, the Bregman loss function $D_\phi : \mathbb{R}^2 \times \mathbb{R}^2 \mapsto \mathbb{R}$ is defined as

$$D_\phi(x, y) = \phi(x) - \phi(y) - \langle x - y, \nabla \phi(y) \rangle.$$

The Bregman loss function class covers common losses such as the squared loss $D_\phi(x, y) := \|x - y\|_2^2$ and Kullback-Liebler (KL) divergence $D_\phi(x, y) = \sum_{j=1}^2 x_j \log(x_j/y_j)$ between two probability vectors $x, y$.

Theorem 4.2 states that the prediction probability $\hat{q}_{\text{ang}}$ generated by the angular predictor uniquely minimizes *any* Bregman loss against the ground-truth probability vector $\hat{q}_\star$ within the class of $q_F$ for any $F$. We defer the proof of the Theorem below to Section B.

**Theorem 4.2** (Optimality of angular predictor). *Let $\phi : \mathbb{R}^2 \mapsto \mathbb{R}$ be any strictly convex differentiable function, and let $D_\phi$ be the corresponding Bregman loss function. Let $\mathbb{E}_{x_{\text{new}}}[\phi(q_\star)]$ be finite. Then, the expected Bregman loss $\mathbb{E}_{x_{\text{new}}}[D_\phi(q_\star, \hat{q}_F)]$ admits a unique minimizer (up to a.s. equivalence) among all $q_F, \forall F \in \mathcal{F} := \{f : \mathbb{R} \to [0, 1]\}$. Let this minimizer be $F_\star = \arg\min_{F \in \mathcal{F}} \mathbb{E}_{x_{\text{new}}}[D_\phi(q_\star, \hat{q}_F)]$. Further suppose that the link function $\sigma$ is continuous. We then have that as $n, d \to \infty$, we have*

$$\left\| \hat{q}_{\text{ang}}(\hat{\theta}) - \hat{q}_{F_\star}(\widehat{w}^\top x_{\text{new}}) \right\|_2^2 \to 0$$

*in probability. That is, the label prediction probability vector from angular calibration converges to the optimal label prediction probability vector given by $F_\star$.*

We note that as $\hat{\theta} = \theta_*$, $\hat{q}_{\text{ang}}(\hat{\theta})$ precisely attains the optimal solution $F_*(\widehat{w}^\top x_{\text{new}})$. We defer the technical statement to Theorem B.2 in Section B.

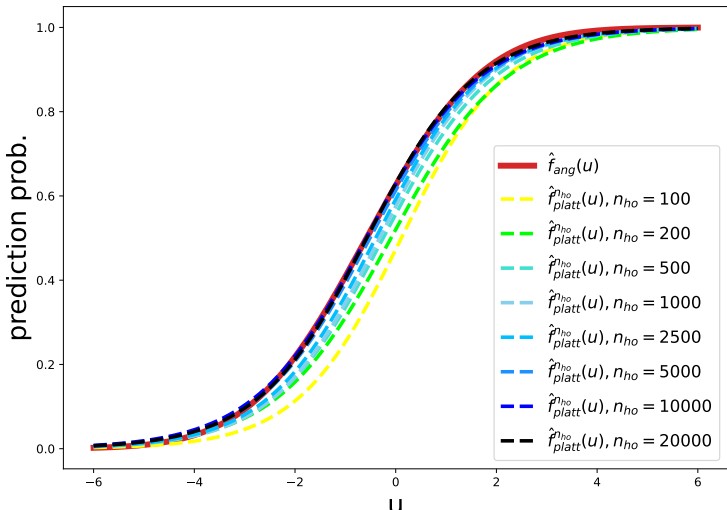

Figure 1: Platt scaling of a logistic ridge predictor converges to angular calibration predictor, as holdout set size increases. The plot is generated with Gaussian data with covariance $\Sigma = \frac{1}{d}\bar{\Sigma}$ where $\bar{\Sigma}_{kl} = 0.5^{|k-l|}, \forall k, l \in \{1, ..., d\}$, sigmoid link function in a data deficient setting where $n = 1000, p = 2000$. See more details in Section 7

## 5 Main Result II: Platt scaling is provably calibrated and Bregman-optimal

Platt scaling is arguably the most widely used calibration method in modern machine learning, yet its theoretical properties in high-dimensional settings remain unexplored. In this section, we identify conditions under which Platt scaling converges to our angular predictor, and is therefore well-calibrated and Bregman-optimal in high dimensions.

Platt scaling finds a mapping $F$ of the prediction logits $\hat{w}^\top x_{\text{new}}$ by minimizing the log-likelihood on a holdout dataset. In this section, we specifically consider the setting where we have a holdout dataset $(x_{\text{ho},i}, y_{\text{ho},i})_{i=1}^{n_{\text{ho}}}$ and the negative log-likelihood

$$\hat{\ell}_{n_{\text{ho}}}(F) := \sum_{i=1}^{n_{\text{ho}}} -y_{\text{ho},i} \log\big(F(\hat{w}^\top x_{\text{ho},i})\big) - (1 - y_{\text{ho},i}) \log\big(1 - F(\hat{w}^\top x_{\text{ho},i})\big). \tag{5}$$

The Platt calibration procedure then searches for a mapping $F$ within some hypothesis class $\mathcal{F}_{\text{platt}}$ that minimizes the negative log-likelihood. Elementary asymptotic theory then shows that as $n_{\text{ho}} \to \infty$ (i.e. the holdout set is sufficiently large), $\hat{L}(\theta)$ in (5) converges to the population loss

$$\ell^\star(F) = \mathbb{E}_{x_{\text{new}}}\left[D_{\text{KL}}\left(\begin{pmatrix} \sigma(w_\star^\top x_{\text{new}}) \\ 1 - \sigma(w_\star^\top x_{\text{new}}) \end{pmatrix} \middle\| \begin{pmatrix} F(\hat{w}^\top x_{\text{new}}) \\ 1 - F(\hat{w}^\top x_{\text{new}}) \end{pmatrix}\right)\right] \tag{6}$$

almost surely up to a constant. This is exactly the argument of the Bregman loss $\mathbb{E}_{x_{\text{new}}}\left[D_\phi(q_\star, \hat{q}_F)\right]$ from Theorem 4.2 for $\phi$ specialized as the negative Shannon entropy. That is, such calibration procedures are essentially trying to optimizing the Bregman loss but within the restricted hypothesis class.

This naturally raises the question of whether calibration procedures such as Platt scaling can achieve the optimal Bregman loss, say in the limit of a sufficiently large holdout set. Theorem 5.1 shows that, if $\sigma$ is a probit link function (or is closely approximated by one up to an affine transformation—for instance, $\text{sigmoid}(x) \approx \Phi(\sqrt{\pi/8}\,x)$), and if the negative log-likelihood (5) is minimized over the hypothesis class of the form $\sigma(Au + B), A, B \in \mathbb{R}$ then the resulting Platt-scaled predictor converges to our angular predictor as $n_{\text{ho}} \to \infty$. Combining this connection with our results for the predictor $\hat{f}_{\text{ang}}(u; \theta_*)$ which is our angular predictor $\hat{f}_{\text{ang}}(u; \hat{\theta})$ with $\hat{\theta} = \theta_*$ exactly. As mentioned previously (see also Theorem A.1 and Theorem B.2 in Appendix), $\hat{f}_{\text{ang}}(u; \theta_*)$ is exactly calibrated and Bregman-optimal, which shows that Platt scaling is both provably calibrated and Bregman optimal. This offers

the first formal high-dimensional guarantees of this kind for the widely well-known Platt scaling procedure. We defer the proof of the Theorem below to Section C.

**Theorem 5.1.** *Consider the predictor $\hat{f}_{\text{platt}}^{n_{\text{ho}}}(u)$ calibrated by the Platt scaling procedure, that is,*

$$\hat{f}_{\text{platt}}^{n_{\text{ho}}}(\widehat{w}^\top x_{\text{new}}) = \sigma(\hat{A}^{n_{\text{ho}}} \cdot \widehat{w}^\top x_{\text{new}} + \hat{B}^{n_{\text{ho}}}),$$

$$\text{with } \hat{A}^{n_{\text{ho}}}, \hat{B}^{n_{\text{ho}}} = \underset{(A,B)\in\mathcal{H}}{\operatorname{argmin}} \hat{\ell}_{n_{\text{ho}}}(u \mapsto \sigma(Au + B)) \tag{7}$$

*for $\hat{\ell}_{n_{\text{ho}}}(\cdot)$ defined in (5). If the link function $\sigma$ satisfies $\sigma(x) = \Phi(a \cdot x + b)$ for some $a \in \mathbb{R}\backslash\{0\}, b \in \mathbb{R}$ and the point $(A_*, B_*)$ defined in (9) is contained in a compact subset $\mathcal{H} \subset \mathbb{R}^2$, the angular predictor defined in (3) satisfies*

$$\hat{f}_{\text{ang}}(u; \theta_*) = \sigma(A_* \cdot u + B_*) \in \mathcal{F}_{\text{platt}}, \tag{8}$$

*where*

$$A_* = \frac{\cos(\theta_*)}{\|\hat{w}\|_\Sigma \sqrt{1 + a^2 \sin^2(\theta_*)}}, \qquad B_* = \frac{b}{a}\left(\frac{1}{\sqrt{1 + a^2 \sin^2(\theta_*)}} - 1\right) \tag{9}$$

*and $\mathcal{F}_{\text{platt}} = \{u \mapsto \sigma(Au + B) : A, B \in \mathbb{R}\}$. Moreover, as $n_{\text{ho}} \to \infty$, we have that $\hat{A}^{n_{\text{ho}}} \to A_*, \hat{B}^{n_{\text{ho}}} \to B_*$ in probability and*

$$\sup_{u \in \mathbb{R}} \left| \hat{f}_{\text{platt}}^{n_{\text{ho}}}(u) - \hat{f}_{\text{ang}}(u; \theta_*) \right| \to 0 \tag{10}$$

*in probability. Here, in-probability convergence is with respect to the randomness of $\{(x_{\text{ho},i}, y_{\text{ho},i})_{i=1}^{n_{\text{ho}}}\}$.*

To be clear, the above theorem considers the asymptotics in the holdout set $n_{\text{ho}}$ for a fixed sample size and dimension $n, d$ of the training dataset. We illustrate Theorem 5.1 in Figure 1 (see Section 7 for detailed settings) where the solid red line plots our angular predictor $u \mapsto \hat{f}_{\text{ang}}(u)$ defined in (3) and the dashed lines plot the predictor $u \mapsto \hat{f}_{\text{platt}}^{n_{\text{ho}}}(u)$ calibrated by Platt scaling on increasingly large holdout sets $n_{\text{ho}}$. We observe that the Platt scaling predictors indeed converge to our angular predictor as the holdout set sizes $n_{\text{ho}}$ increase.

## 6 Consistent angle estimation

Observe that the angular predictor $\hat{f}_{\text{ang}}$ depends on the unobserved quantity $\langle w_\star, \widehat{w} \rangle_\Sigma$. Using recent advancements from Equation (11) we are able to provide a consistent estimator for this quantity. For simplicity, we outline the estimation procedure here for a twice-differentiable loss function $\ell$ and strongly convex, twice-differentiable penalty $g$; analogous results for unregularized M-estimation and other losses/penalties found in [8]. We note that this result is part of a long line of development in observable estimation of unknown quantities in high dimensions [38, 9, 10, 18, 54].

A data driven estimator for $\langle w_\star, \widehat{w} \rangle_\Sigma^2$ proposed in [8] is:

$$\hat{a}_*^2 = \frac{\left(\frac{\hat{v}}{n}\|X\widehat{w} - \hat{\gamma}\hat{\psi}\|^2 + \frac{1}{n}\hat{\psi}^\top X\widehat{w} - \hat{\gamma}\hat{r}^2\right)^2}{\frac{1}{n^2}\left\|\Sigma^{-\frac{1}{2}}X^\top\hat{\psi}\right\|^2 + \frac{2\hat{v}}{n}\hat{\psi}^\top X\widehat{w} + \frac{\hat{v}^2}{n}\|X\widehat{w} - \hat{\gamma}\hat{\psi}\|^2 - \frac{d}{n}\hat{r}^2} \tag{11}$$

where $\hat{\psi} \in \mathbb{R}^n$ is the vector with components $\hat{\psi}_i = -\ell_i'\left(x_i^\top \widehat{w}\right), \hat{v} = \frac{1}{n}\operatorname{Tr}\left(D - DX\hat{H}X^TD\right), \hat{\gamma} = \operatorname{Tr}\left(X\hat{H}X^\top D\right)$ for $D = \operatorname{diag}\left(\ell_y''(X\widehat{w})\right)$ and $\hat{H} = \left(X^\top DX + n\nabla^2 g(\widehat{w})\right)^{-1}$ and $\hat{r} = (\frac{\|\hat{\psi}\|^2}{n})^{1/2}$. It can be shown that $\left|\hat{a}_*^2 - \langle w_\star, \widehat{w} \rangle_\Sigma^2\right| \to 0$ in suitable high-dimensional sense. We refer the technical statement to Theorem D.1 in Section D.

To estimate $\langle w_\star, \widehat{w} \rangle_\Sigma$, we also need to estimate its sign. We require reserving a constant fraction $n_{\text{ho}} = \alpha \cdot n$ of the $n$ training data for the sign estimator,

$$\widehat{\text{sgn}} := \text{sign}\left(\sum_{i=1}^{n_{\text{ho}}} \widehat{w}^\top x_i^{\text{ho}} \cdot y_i^{\text{ho}}\right). \tag{12}$$

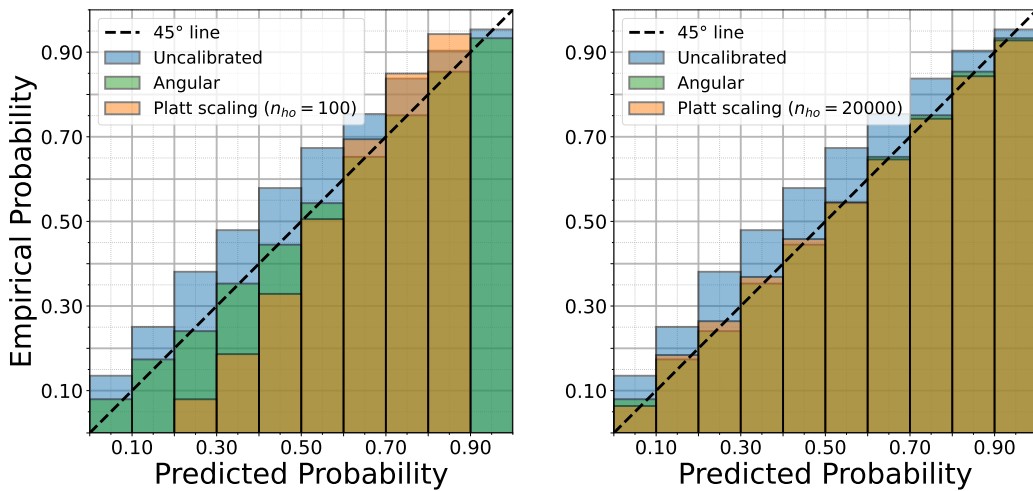

Figure 2: Reliability plots for angular calibration and Platt scaling of a logistic ridge predictor. Left panel uses a small holdout set for Platt scaling with $n_{\text{ho}} = 100$; Right panel uses a large holdout set with $n_{\text{ho}} = 2000$. The plot is generated with Gaussian data with covariance $\Sigma = \frac{1}{d}\bar{\Sigma}$ where $\bar{\Sigma}_{kl} = 0.5^{|k-l|}, \forall k, l \in \{1, ..., d\}$, sigmoid link function in a data deficient setting where $n = 1000, p = 2000$. See Section 7 for more details.

It can be shown that the probability of wrong sign identification using $\widehat{\text{sgn}}$ decreases exponentially with $n_{\text{ho}}$. We defer the proof to Section E.

**Proposition 6.1.** *Suppose that $\sigma'(x)$ is well-defined and non-negative almost everywhere and $\sigma'(x) > 0$ on a set with non-zero Lebesgue measure. We then have for some absolute constant $c > 0$,*

$$\mathbb{P}_{\text{ho}}\left(\widehat{\text{sgn}} = \text{sign}\left(\langle w_\star, \widehat{w}\rangle_\Sigma\right)\right) \geq 1 - 2\exp\left(-cn_{\text{ho}}\left(\cos\left(\theta_*\right)\cdot\mathbb{E}\sigma'(Z)\right)^2\right)$$

*where $Z \sim N(0, 1)$ and $\mathbb{P}_{\text{ho}}$ is with respect to the randomness in $\left(y_i^{\text{ho}}, x_i^{\text{ho}}\right)_{i=1}^{n_{\text{ho}}}$.*

Plugging (11) and (12) into (2), we have the following estimator $\hat{\theta}$ for $\theta_*$ in (2)

$$\hat{\theta} := \arccos\left(\|\widehat{w}\|_\Sigma^{-1}\widehat{\text{sgn}}\cdot\sqrt{\hat{a}_*^2}\right). \tag{13}$$

Theorem 6.2 below shows that $\hat{\theta}$ is consistent.

**Corollary 6.2.** *Under the assumption of Proposition 6.1 and Theorem D.1, as $n, d \to \infty$, we have that $|\hat{\theta} - \theta_*| \to 0$ in probability where $n_{\text{ho}} = \alpha \cdot n$ for a fixed constant $\alpha > 0$.*

## 7  Numerical experiments

### 7.1  Simulations

This section presents a simple simulation to demonstrate results in Section 6. We generate i.i.d. samples $x_i \stackrel{\text{iid}}{\sim} N(0, \Sigma), i = 1, ..., n$ where $\Sigma = \frac{1}{d}\bar{\Sigma}$ and $\bar{\Sigma}_{kl} = 0.5^{|k-l|}, \forall k, l \in \{1, ..., d\}$; we also generate labels from (1) with $\sigma(u) = \text{sigmoid}(3u + 1) = 1/(1 + \exp(-(3u + 1)))$ and $w_\star \sim N(0, I_d)$ (normalized to $\|w_\star\|_\Sigma = 1$). We consider the case of ridge logistic regression with $\ell_{y_i}(w^\top x) = -y_i\log(\hat{p}_w(x_i)) - (1 - y_i)\log(1 - \hat{p}_w(x_i))$ with $\hat{p}_w(x_i) = \frac{1}{1+\exp(-x_i w)}$ and $g(w) = \frac{\lambda}{2d}\|w\|_2^2$ with $\lambda = 0.5$. We assume that we are in a data deficient setting where $n = 1000, p = 2000$.

The realizability plots in Figure 2 are generated from a test set of size $n_{\text{test}} = 20000$. To produce these plots, we bin the predicted probabilities for label 1 (on the x-axis) and then compute the average of the observed label within each bin (on the y-axis). Perfect calibration would align the binned

Table 1: **ECE on pretrained feature extractors.** "Uncal." = uncalibrated; "Angular" = our method; "Platt/Iso" use $n_{\text{ho}} \in \{100, 500\}$ labeled hold-out points.

| Model–Dataset | Uncal. | Angular | Platt 100 | Iso 100 | Platt 500 | Iso 500 |
|---|---|---|---|---|---|---|
| ResNet-34–CIFAR-10 | 0.1236 | 0.0199 | 0.0561 | 0.0484 | 0.0259 | 0.0298 |
| MiniLM–20 Newsgroups | 0.1392 | 0.0249 | 0.0931 | 0.1107 | 0.0679 | 0.0813 |
| ChemBERTa–Tox21 | 0.1389 | 0.0132 | 0.0236 | 0.0497 | 0.0175 | 0.0293 |

points with the 45° line. In the left and right panels, Platt scaling is derived using holdout sets of $n_{\text{ho}} = 100$ and $n_{\text{ho}} = 20000$, respectively, whereas both the uncalibrated predictor and the angular predictor remain unchanged across the two panels.

From the reliability plots, we see that the uncalibrated predictor (blue) is poorly calibrated, while, as expected, the angular predictor (green) shows good calibration. Here, we have used the angle estimator (11) and the sign estimator (12) to estimate the value of $\langle w_\star, \widehat{w} \rangle_\Sigma$. The estimated value for $\langle w_\star, \widehat{w} \rangle_\Sigma$ is 0.4356 while the true value is 0.4526. We also ran 5000 Monte Carlo trials where we found the probability of incorrect sign estimation to be 0.89% with a holdout set of size $n_{\text{ho}} = 100$.

In contrast, the left panel of Figure 2 shows that Platt scaling (orange) with a holdout set size of $n_{\text{ho}} = 100$ fails to properly calibrate. However, when the holdout set size is increased to $n_{\text{ho}} = 20000$, Platt scaling also calibrates correctly. When $n_{\text{ho}} = 20000$, the predictor calibrated from Platt scaling is found to be (using scikit-learn package's CalibratedClassifierCV routine [68])

$$\hat{f}_{\text{platt}}^{n_{\text{ho}}}(\widehat{w}^\top x_{\text{new}}) = \sigma(\hat{A}^{n_{\text{ho}}} \cdot \widehat{w}^\top x_{\text{new}} + \hat{B}^{n_{\text{ho}}}), \text{ with } \hat{A}^{n_{\text{ho}}} = 0.3396, \hat{B}^{n_{\text{ho}}} = -0.1521.$$

We now check if $\hat{A}^{n_{\text{ho}}}$ and $\hat{B}^{n_{\text{ho}}}$ are indeed close to $A_*, B_*$ as claimed in Theorem 5.1. Note that even though the link function is not strictly a probit function as required by Theorem 5.1, by approximating $\text{sigmoid}(u) \approx \Phi(\sqrt{\pi/8} \cdot u)$, we have $\sigma(u) \approx \Phi(\sqrt{\pi/8}(3u + 1))$. We then obtain $A_* = 0.2991, B_* = -0.1597$ from (9) setting $a = 3\sqrt{\pi/8}, b = \sqrt{\pi/8}$, which are indeed quite close to $\hat{A}^{n_{\text{ho}}} = 0.3396, \hat{B}^{n_{\text{ho}}} = -0.1521$.

## 7.2 Semi-real experiments

We assess angular calibration on semi-real tasks that keep real-data covariates but simulate labels from the known generative model. We maintain settings in Section 7.1 but replace data covariates with: (i) final-layer logits of pretrained deep networks; and (ii) classic UCI benchmarks.

We found that (11) plug-in estimator for $\langle w^\star, \hat{w} \rangle$ is unstable on these real datasets. This is a known issue for estimators like (11) that are based on Wigner-type random-matrix-theoretic assumptions. Modifying these estimator are an active research area [54, 57]. To isolate calibration effects, we simulate $w^\star$ and labels, using the true angle.

Each dataset is split into training set $n$, a large unlabeled pool $n_{\text{cov}} \gg n_{\text{train}}$ for covariance estimation, and $n_{\text{test}}$. We report Expected Calibration Error (ECE; lower is better) [29]. Post-hoc baselines use a labeled hold-out of size $n_{\text{ho}} \in \{100, 500\}$ ("Platt 100/500" and "Iso 100/500").

**Pretrained representations.** We fit a linear head on frozen embeddings and calibrate the resulting logits: ResNet-34 (ImageNet-1K pretrain) on CIFAR-10 [35, 24, 46], MiniLM sentence embeddings on 20 Newsgroups [85, 72, 51], and ChemBERTa on Tox21 from MoleculeNet [20, 87]. We have $n \times d = (300 \times 512)/(800 \times 384)/(800 \times 768)$, $n_{\text{cov}} = 30,000/3,000/3,000$ and $n_{\text{test}} = 10,000/1,000/500$ for CIFAR-10 / 20NG / Tox21. The results are reported in Table 1; the reliability plots are deferred to Section H.2.

**UCI benchmarks.** On Communities & Crime, Splice-junction Gene Sequences, and Madelon [71, 1, 32], we train linear predictors on raw covariates with $n \times d = (200 \times 100)/(100 \times 180)/(300 \times 500)$, $n_{\text{cov}} = 700/2000/900$, and $n_{\text{test}} = 593/900/900$ for Communities & Crime / Splice-junction / Madelon. The results are reported in Table 2; the reliability plots are deferred to Section H.2.

Table 2: **ECE on UCI datasets.** Conventions as in Table 1.

| Dataset | Uncal. | Angular | Platt 100 | Iso 100 | Platt 500 | Iso 500 |
|---|---|---|---|---|---|---|
| Communities & Crime | 0.1700 | 0.0296 | 0.0620 | 0.0661 | 0.0262 | 0.0436 |
| Splice-junction Gene Seq | 0.1262 | 0.0208 | 0.0568 | 0.1041 | 0.0556 | 0.0879 |
| Madelon | 0.1473 | 0.0267 | 0.0721 | 0.0719 | 0.0299 | 0.0624 |

## 8 Extensions and future directions

We derived our theoretical results assuming that the covariates are Gaussian—although at first pass this might appear stylistic, recent universality results [37, 56, 33, 26, 49, 62] demonstrate that these results should continue to hold as long as the covariates have sufficiently light tails. We demonstrate this with further experiments. In Section H.2 and Figure 4, we reproduce Figure 1 and 2 with non-Gaussian design matrices (iid Rademacher and uniform entries respectively) where we observe that our results continue to be accurate. Establishing such universality formally should be an interesting avenue for future work—we include an informal discussion here. Consider a general setting where $x_i \overset{\mathrm{d}}{=} \Sigma^{1/2} z_i$, where $z_i$ has iid entries with zero mean, unit variance and finite moments, and $\Sigma = p^{-1}\bar{\Sigma}$ where $\bar{\Sigma}$ has bounded condition number. Denote $\widetilde{w} = \bar{\Sigma}^{1/2}\widehat{w}, \widetilde{w}_\star = \bar{\Sigma}^{1/2} w^\star$ and $x_{\mathrm{new}}, z_{\mathrm{new}}$ to be observations at test time with the same distribution as $x_i, z_i$ respectively. If we could apply the multivariate CLT [11], we would obtain

$$\left( p^{-1/2} \sum_{i=1}^{p} \widetilde{w}_{\star,i} z_{\mathrm{new},i}, p^{-1/2} \sum_{i=1}^{p} \widetilde{w}_i z_{\mathrm{new},i} \right) \Rightarrow (Z_1, Z_2), \tag{14}$$

where $(Z_1, Z_2) \sim N(0, L)$ for some positive definite covariance matrix $L \in \mathbb{R}^{2\times 2}$. To apply the multivariate CLT, we require to check the following moment condition (c.f. [11])

$$p^{-3/2} \sum_{i=1}^{p} \mathbb{E}\left[ z_{\mathrm{new},i}^{\frac{3}{2}} \right] \left[ (\widetilde{w}_{\star,i}, \widetilde{w}_i) \begin{pmatrix} \|\widehat{w}\|_\Sigma^2 & \langle \widehat{w}, w_\star \rangle_\Sigma \\ \langle \widehat{w}, w_\star \rangle_\Sigma & \|w_\star\|_\Sigma^2 \end{pmatrix}^{-1} \begin{pmatrix} \widetilde{w}_{\star,i} \\ \widetilde{w}_i \end{pmatrix} \right]^{\frac{3}{2}}$$
$$\leq p^{-1/2} \mathbb{E}\left[ z_{\mathrm{new},i}^{\frac{3}{2}} \right] \sigma_{\min}^{-1}(L) \left( \frac{1}{p} \sum_{i=1}^{p} |\widetilde{w}_{\star,i}|^3 + \frac{1}{p} \sum_{i=1}^{p} |\widetilde{w}_i|^3 \right) = o\left( \frac{1}{\sqrt{p}} \right).$$

We claim that the above holds almost surely for sufficiently large $p$, if we have constants $W_1, W_2 > 0$ for which the following holds

$$\begin{pmatrix} \|\widehat{w}\|_\Sigma^2 & \langle \widehat{w}, w_\star \rangle_\Sigma \\ \langle \widehat{w}, w_\star \rangle_\Sigma & \|w_\star\|_\Sigma^2 \end{pmatrix} \to L, \quad \left( \frac{1}{p} \sum_{i=1}^{p} |\widetilde{w}_{\star,i}|^3, \frac{1}{p} \sum_{i=1}^{p} |\widetilde{w}_i|^3 \right) \to (W_1, W_2).$$

Recent universality results for either approximate message passing algorithms [19] or convex gaussian minmax theorems (CGMT) [33] allow one to prove this beyond Gaussian designs. Numerous works have already applied such arguments in the context of other high-dimensional problems [37, 62, 56, 49]. Using (14), the conditional distribution (15) in the proof of Theorem 3.2 can be extended to non-Gaussian, Wigner-type features preconditioned by some known $\Sigma^{1/2}$, thus leading to a proof of Theorem 2 beyond Gaussian designs. In the interest of space, we defer formalizing this to future work.

Finally, we consider binary classification in this work; it would be interesting to extend our results to multi-index models, which includes multi-class classification, additive and interaction models, and two-layer neural networks [82, 88, 15]; see details in Section F. Multi-index model can be defined as follows: for $K \geq 2$ and unobserved indices $W_\star = [w_{\star 1}, \ldots, w_{\star K}] \in \mathbb{R}^{d\times K}$, the true logits and model outputs are $G := W_\star^\top x_{\mathrm{new}} \in \mathbb{R}^K$, $\pi(x_{\mathrm{new}}) = g(G)$ where $g$ is a generalized link (vector- or scalar-valued). We show in Section F that, given an estimator $\widehat{W} = [\widehat{w}_1, \ldots, \widehat{w}_K]$ of $W_\star$, angular predictor in Definition 3.1 may be extended for the multi-index model as follows,

$$\widehat{f}_{\mathrm{ang}}\left( \widehat{W}^\top x_{\mathrm{new}} \right) := \mathbb{E}_Z\left[ g\left( M_\star S + L_\star Z \right) \right]$$

where the matrix quantity $M_\star$ and $L_\star$ depends on cross-index angles $\langle w_{\star k}, \widehat{w}_\ell \rangle_\Sigma$, $\ell, k \in [K]$. Though no estimators are given for these cross-index angles in literature as far as we know, recent theory for multi-index models [82] suggests that analogues of the single-index angle estimators [8] are feasible. We leave this to future works.

## Acknowledgments and Disclosure of Funding

P.S. was funded partially by NSF DMS Award No. 2113426 and NSF CAREER Award No. 2440824.

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

# A  Proof of Theorem 3.2

Before proving Theorem 3.2, we first show that $\hat{f}_{\mathrm{ang}}(\cdot;\theta_*)$ is exactly calibrated.

**Theorem A.1.** *The predictor $\hat{f}_{\mathrm{ang}}$ defined in* (3) *is well-calibrated at all $p \in [0,1]$ when . That is, for any $p \in [0,1]$ and any $d, n \in \mathbb{N}_+$*

$$\Delta_p^{\mathrm{cal}}\left(\hat{f}_{\mathrm{ang}}(\cdot;\theta_*)\right) = p - \mathbb{E}_{x_{\mathrm{new}}}\left[\sigma\left(w_\star^\top x_{\mathrm{new}}\right) \mid \hat{f}_{\mathrm{ang}}\left(\hat{w}^\top x_{\mathrm{new}};\theta_*\right) = p\right] = 0.$$

*Proof of Theorem A.1.* Let us define the following event

$$\mathcal{A} := \left\{\hat{f}_{\mathrm{ang}}\left(w_\star^\top x_{\mathrm{new}};\theta_*\right) = p\right\}.$$

We have

$$\mathbb{E}_{x_{\mathrm{new}}}\left[\sigma\left(w_\star^\top x_{\mathrm{new}}\right) \mid \mathcal{A}\right]$$
$$\overset{(i)}{=} \mathbb{E}_{x_{\mathrm{new}}}\left[\mathbb{E}_{x_{\mathrm{new}}}\left[\sigma\left(w_\star^\top x_{\mathrm{new}}\right) \mid x_{\mathrm{new}}^\top \hat{w}\right] \mid \mathcal{A}\right]$$
$$\overset{(ii)}{=} \mathbb{E}_{x_{\mathrm{new}}}\left[\mathbb{E}_Z\left[\sigma\left(\frac{1}{\|\hat{w}\|_\Sigma} \cdot \cos(\theta_*) \cdot x_{\mathrm{new}}^\top \hat{w} + \sin(\theta_*) \cdot Z\right)\right] \mid \mathcal{A}\right]$$
$$= \mathbb{E}_{x_{\mathrm{new}}}\left[\hat{f}_{\mathrm{ang}}\left(w_\star^\top x_{\mathrm{new}};\theta_*\right) \mid \mathcal{A}\right]$$
$$= p.$$

where (i) follows from tower property of expectation and the fact that $\hat{f}_{\mathrm{ang}}(x_{\mathrm{new}})$ depends on $x_{\mathrm{new}}$ only through $x_{\mathrm{new}}^\top \hat{w}$, (ii) follows from conditional expectation of multivariate Gaussian distribution

$$\left[w_\star^\top x_{\mathrm{new}} \mid x_{\mathrm{new}}^\top \hat{w}\right] \overset{L}{=} \frac{1}{\|\hat{w}\|_\Sigma} \cdot \cos(\theta_*) \cdot x_{\mathrm{new}}^\top \hat{w} + \sin(\theta_*) \cdot Z \tag{15}$$

for some $Z \sim N(0,1)$. $\qquad\square$

*Proof of Theorem 3.2.* Using result from Theorem A.1, it suffices to show that as $|\hat{\theta} - \theta_*| \to 0$ in probability, we have that

$$\left|\Delta_p^{\mathrm{cal}}\left(\hat{f}_{\mathrm{ang}}(\cdot;\theta_*)\right) - \Delta_p^{\mathrm{cal}}\left(\hat{f}_{\mathrm{ang}}(\cdot;\hat{\theta})\right)\right| \to 0. \tag{16}$$

Let us introduce the following notation for the ease of presentation:

$$X := \sigma(w_\star^\top x_{\mathrm{new}}), \quad \hat{Y} = \hat{f}_{\mathrm{ang}}\left(\hat{w}^\top x_{\mathrm{new}};\hat{\theta}\right), \quad Y_* = \hat{f}_{\mathrm{ang}}\left(\hat{w}^\top x_{\mathrm{new}};\theta_*\right).$$

Then, we can write LHS of (16) as

$$\left|\mathbb{E}[X \mid \hat{Y} = p] - \mathbb{E}[X \mid Y_\star = p]\right| = \left|\frac{1}{f_{\hat{Y}}(p)}\int_0^1 x f_{X,\hat{Y}}(x,p)\mathrm{d}x - \frac{1}{f_{Y_\star}(p)}\int_0^1 x f_{X,Y_\star}(x,p)\mathrm{d}x\right|$$

where $f_{\hat{Y}}, f_{Y_*}, f_{X,Y_\star}, f_{X,\hat{Y}}$ are the distribution density functions of $\hat{Y}, Y_*$ and joint density functions of $(X, Y_\star)$ and $(X, \hat{Y})$. We now show that the RHS of the above converges to 0. Firstly, $|\frac{1}{f_{Y_\star}(p)} - \frac{1}{f_{\hat{Y}}(p)}| \to 0$ because $|\hat{Y} - Y_\star| \to 0$ in probability (and thus in distribution) by continuous mapping theorem. Secondly,

$$\left|\int_0^1 x f_{X,\hat{Y}}(x,p)\mathrm{d}x - \int_0^1 x f_{X,Y_\star}(x,p)\mathrm{d}x\right| \to 0$$

by bounded convergence theorem and the fact that $(X, \hat{Y})$ converges to $(X, Y_*)$ jointly. We conclude the proof. $\qquad\square$

# B  Proof of Theorem 4.2

We first state a result from [4] for general random variables.

**Proposition B.1** (Theorem 1, [4]). *Let $\phi : \mathbb{R}^d \mapsto \mathbb{R}$ be a strictly convex differentiable function, and let $D_\phi$ be the corresponding Bregman loss function. Let $X$ be an arbitrary random variable taking values in $\mathbb{R}^d$ for which both $\mathbb{E}[X]$ and $\mathbb{E}[\phi(X)]$ are finite. Then, among all functions of $Z$, the conditional expectation is the unique minimizer (up to a.s. equivalence) of the expected Bregman loss, i.e.,*

$$\arg \min_{Y \in \sigma(Z)} \mathbb{E}\left[D_\phi(X, Y)\right] = \mathbb{E}[X \mid Z].$$

Using the above results, we show that angular calibration with $\hat{\theta} = \theta_*$ minimizes Bregman divergence to true label distribution among predictors of the form $F(\widehat{w}^\top x_{\text{new}})$.

**Theorem B.2** (Optimality of angular predictor). *Let $\phi : \mathbb{R}^2 \mapsto \mathbb{R}$ be any strictly convex differentiable function, and let $D_\phi$ be the corresponding Bregman loss function. Let $\mathbb{E}_{x_{\text{new}}}[\phi(q_\star)]$ be finite. Then, the expected Bregman loss $\mathbb{E}_{x_{\text{new}}}[D_\phi(q_\star, \hat{q}_F)]$ admits a unique minimizer (up to a.s. equivalence) among all $q_F, \forall F \in \mathcal{F} := \{f : \mathbb{R} \to [0, 1]\}$. Let this minimizer be $F_\star = \arg\min_{F \in \mathcal{F}} \mathbb{E}_{x_{\text{new}}}[D_\phi(q_\star, \hat{q}_F)]$. We then have that almost surely*

$$\hat{q}_{\text{ang}}(\theta_*) = F_*(\widehat{w}^\top x_{\text{new}})$$

*where $\hat{q}_{\text{ang}}(\theta_*)$ is the label prediction probability vector by angular calibration given in (4) with $\hat{\theta}$ replaced by $\theta_*$.*

*Proof of Theorem B.2.* Firstly, we set $X, Y, Z$ in Theorem B.1 as

$$X \leftarrow \begin{pmatrix} \sigma(w_\star^\top x_{\text{new}}) \\ 1 - \sigma(w_\star^\top x_{\text{new}}) \end{pmatrix}, \quad Y \leftarrow \begin{pmatrix} F(\widehat{w}^\top x_{\text{new}}) \\ 1 - F(\widehat{w}^\top x_{\text{new}}) \end{pmatrix}, \quad Z \leftarrow \widehat{w}^\top x_{\text{new}}.$$

The result then follows from Theorem B.1 and the following

$$\mathbb{E}[X \mid Z] = \mathbb{E}\left[\begin{pmatrix} \sigma(w_\star^\top x_{\text{new}}) \\ 1 - \sigma(w_\star^\top x_{\text{new}}) \end{pmatrix} \mid x_{\text{new}}^\top \widehat{w}\right] = \begin{pmatrix} \hat{f}_{\text{ang}}(x_{\text{new}}) \\ 1 - \hat{f}_{\text{ang}}(x_{\text{new}}) \end{pmatrix}$$

where we used (15) and (3) for the last equality. $\quad\square$

Now we are ready to state proof of Theorem 4.2.

*Proof.* Using result from Theorem B.2, it suffices to show that

$$\left\| \hat{q}_{\text{ang}}(\hat{\theta}) - \hat{q}_{\text{ang}}(\theta_*) \right\|_2 \to 0$$

in probability. This is an immediate consequence of the continuous mapping theorem under the assumption $\sigma$ is continuous. $\quad\square$

# C  Proof of Theorem 5.1

Before proving Theorem 5.1, we first state two classic analysis results that we will later use.

**Proposition C.1** (Theorem 5.7, [83]). *Let $\ell_n$ be random functions on $\mathcal{H}$, $\ell^\star$ be a fixed function on $\mathcal{H}$, and $\theta^\star \in \mathcal{H}$ such that (i) uniform convergence of $n^{-1}\ell_n$ to $\ell^\star$ holds:*

$$\sup_{\theta \in \mathcal{H}} \left| \frac{1}{n}\ell_n(\theta) - \ell^\star(\theta) \right| \xrightarrow[n \to \infty]{\text{inprob.}} 0,$$

*(ii) the mode of $\ell^\star$ is well-separated, i.e for all $\varepsilon > 0$,*

$$\sup_{\theta \in \mathcal{H}: d(\theta, \theta^\star) \geq \varepsilon} \ell^\star(\theta) < \ell^\star(\theta^\star)$$

*Then any sequence $\hat{\theta}_n$ maximizing $\ell_n$ converges in probability to $\theta^\star$.*

**Proposition C.2** (Theorem 10.8, [73]). *Let $C$ be a relatively open convex set, and let $f_1, f_2, \ldots,$ be a sequence of finite convex functions on $C$. Suppose that the sequence converges pointwise on a dense subset of $C$, i.e. that there exists a subset $C'$ of $C$ such that its closure satisfies $\mathrm{cl}\,C' \supset C$ and, for each $x \in C'$, the limit of $f_1(x), f_2(x), \ldots,$ exists and is finite. The limit then exists for every $x \in C$, and the function $f$, where*

$$f(x) = \lim_{i \to \infty} f_i(x)$$

*is finite and convex on $C$. Moreover the sequence $f_1, f_2, \ldots,$ converges to $f$ uniformly on each closed bounded subset of $C$.*

Now we are ready to prove Theorem 5.1.

*Proof of Theorem 5.1.* (8) is obtained from applying the well-known identity below for probit function $\Phi(\cdot)$

$$\mathbb{E}\Phi(\mu + \sigma \cdot Z) = \Phi\left(\frac{\mu}{\sqrt{1 + \sigma^2}}\right), \qquad Z \sim N(0, 1)$$

to (3).

To prove that $\hat{A}^{n_{\mathrm{ho}}} \to A_*, \hat{B}^{n_{\mathrm{ho}}} \to B_*$ as $n_{\mathrm{ho}} \to \infty$, we would like to apply (C.1) by setting $n \leftarrow n_{\mathrm{ho}}, \theta \leftarrow \{A, B\}, \theta^\star \leftarrow \{A_*, B_*\}$,

$$\ell_n(\theta) \leftarrow \ell_{n_{\mathrm{ho}}}(A, B) := \sum_{i=1}^{n_{\mathrm{ho}}} -y_{\mathrm{ho},i} \log\big(F_{A,B}(\widehat{w}^\top x_{\mathrm{ho},i})\big) - (1 - y_{\mathrm{ho},i}) \log\big(1 - F_{A,B}(\widehat{w}^\top x_{\mathrm{ho},i})\big),$$
(17)

and

$$\ell^\star(\theta) \leftarrow \ell^\star(A, B) := \mathbb{E}_{x_{\mathrm{new}}} \Bigg[ -\sigma\left(w_\star^\top x_{\mathrm{new}}\right) \log\left(F_{A,B}\left(\widehat{w}^\top x_{\mathrm{new}}\right)\right)$$
$$- \left(1 - \sigma\left(w_\star^\top x_{\mathrm{new}}\right)\right) \log\left(1 - F_{A,B}\left(\widehat{w}^\top x_{\mathrm{new}}\right)\right) \Bigg].$$
(18)

where we used the notation

$$F_{A,B}(u) := \sigma(Au + B) = \Phi(a(Au + B) + b).$$

Here, RHS of (18) is up to an affine transform of the KL divergence (6) and, therefore, it follows from (8) and (**??**) that its minimizer is indeed $A_*, B_*$.

To verify condition (i) of Theorem C.1, we first note that (17) converges to (18) point-wise in probability following from law of large number theorem. Furthermore, we note that the functions

$$f_1(u) := -\log(\Phi(u)), \qquad f_2(u) := -\log(1 - \Phi(u))$$

are both strictly convex functions. To see this, one can show second derivatives are positive for all $u \in \mathbb{R}$ by utilizing the two elementary inequalities: $u\Phi(u) + \Phi'(u) > 0, u\Phi(u) + \Phi'(u) > u$. It then follows that $\ell_{n_{\mathrm{ho}}}(A, B)$ is a convex function of $(A, B)$ on the domain $\mathbb{R}^2$. Uniform convergence on $\mathcal{H}$ is then an application of Theorem C.2.

To verify condition (ii) of Theorem C.1, we only need to show that $\ell^\star(A, B)$ is a strictly convex function in $(A, B)$. Let us write the inside of the expectation of (18) as

$$f(A, B) = \sigma(H)f_1(aA \cdot H + aB + b) + (1 - \sigma(H))f_2(aA \cdot H + aB + b)$$

where $H := w_\star^\top x_{\mathrm{new}} \sim N(0, 1)$. Then, we have that

$$\nabla^2 f(A, B) = \big(\sigma(H) \cdot f_1''(aA \cdot H + aB + b) \cdot + (1 - \sigma(H))f_2''(aA \cdot H + aB + b)\big) \cdot \begin{pmatrix} a^2 H^2 & a^2 H \\ a^2 H & a^2 \end{pmatrix}$$

which is positive-definite almost surely when $a \neq 0$. Hence, we have that almost surely

$$f\left(t(A_1, B_1) + (1 - t)(A_1, B_1)\right) < tf\left((A_1, B_1)\right) + (1 - t)f\left((A_1, B_1)\right)$$

which implies that

$$\mathbb{E}_{x_{\mathrm{new}}} f\left(t(A_1, B_1) + (1 - t)(A_1, B_1)\right) < t\mathbb{E}_{x_{\mathrm{new}}} f\left((A_1, B_1)\right) + (1 - t)\mathbb{E}_{x_{\mathrm{new}}} f\left((A_1, B_1)\right).$$

The claim that $\ell^\star(A, B)$ is strictly convex follows.

It then follows from Theorem C.1 that $\hat{A}^{n_{\text{ho}}} \to A_*, \hat{B}^{n_{\text{ho}}} \to B_*$ in probability as $n_{\text{ho}} \to \infty$. The uniform convergence $\hat{f}_{\text{platt}}^{n_{\text{ho}}}(u) \to \hat{f}_{\text{ang}}(u)$ follows immediately.

$\square$

## D  Inner product estimation

We restate the following results from Theorem 4.4, [8]. We note that the quantity $\frac{\hat{r}^4}{\hat{v}^2\hat{t}^2}$ in the error bound is observable and is typically of constant order in the proportional regime. See [8] for details.

**Theorem D.1.** *Suppose $\ell$ is continuously differentiable and $g$ is strongly convex and twice differential penalty function. Assume also that $\frac{1}{2\delta} \leq \frac{d}{n} \leq \frac{1}{\delta}$ for some $\delta > 0$, for arbitrarily large probability $1 - \delta$, the following holds*

$$\mathbb{E}\left|\hat{a}_*^2 - \langle w_\star, \widehat{w}\rangle_\Sigma^2\right| \leq \frac{C\hat{r}^4}{\hat{v}^2\hat{t}^2} \cdot n^{-1/2}$$

*where $C$ is a constant depending only on $g$ and $\delta$.*

## E  Sign estimation

*Proof of Theorem 6.1.* With respect to randomness in validation dataset (that is, $\widehat{w}$ is treated as deterministic), we have that $\widehat{w}^\top x_i^{\text{ho}} \cdot y_i^{\text{ho}}$ are iid across $i$ and satisfies that

$$\widehat{w}^\top x_i^{\text{ho}} \cdot y_i^{\text{ho}} \stackrel{L}{=} \widehat{w}^\top x_i^{\text{ho}} \cdot \text{Bern}_i\left(\sigma\left(\frac{\langle w_\star, \widehat{w}\rangle_\Sigma}{\widehat{w}^\top\Sigma\widehat{w}}\widehat{w}^\top x_i^{\text{ho}} + \sin\left(\theta_*\right)\cdot Z_i\right)\right)$$

$$\stackrel{L}{=} \|\widehat{w}\|_\Sigma U_i \cdot \text{Bern}_i\left(\sigma\left(\cos\left(\theta_*\right)U_i + \sin\left(\theta_*\right)\cdot Z_i\right)\right) = H_i$$

where $Z_i \stackrel{iid}{\sim} N(0,1)$ and we have used $w^\top x_i^{\text{ho}} \stackrel{L}{=} \|\widehat{w}\|_\Sigma \cdot U_i$ for $U_i \stackrel{iid}{\sim} N(0,1)$. It follows from Gaussian integration by parts that

$$\mathbb{E}H_i = \langle w_\star, \widehat{w}\rangle_\Sigma \cdot \mathbb{E}\sigma'\left(\cos\left(\theta_*\right)U_i + \sin\left(\theta_*\right)\cdot Z_i\right) = \langle w_\star, \widehat{w}\rangle_\Sigma \mathbb{E}\sigma'(Z).$$

Meanwhile, $H_i$ is subGaussian with subGaussian norm

$$\|H_i\|_{\psi_2}^2 \leq \|\widehat{w}\|_\Sigma^2 \|U_i\|_{\psi_2}^2 \leq 3\|\widehat{w}\|_\Sigma^2.$$

By theorem assumption, $\mathbb{E}\sigma'(Z) > 0$ and

$$\text{sign}\left(\langle w_\star, \widehat{w}\rangle_\Sigma \mathbb{E}\sigma'(Z)\right) = \text{sign}\left(\langle w_\star, \widehat{w}\rangle_\Sigma\right)$$

So the sign identification of $\widehat{\text{sgn}}$ is correct if the following event holds

$$\left|\frac{1}{n_{\text{ho}}}\sum_{i=1}^{n_{\text{ho}}} w^\top x_i \cdot y_i - \langle w_\star, \widehat{w}\rangle_\Sigma \mathbb{E}\sigma'(Z)\right| < \left|\langle w_\star, \widehat{w}\rangle_\Sigma \mathbb{E}\sigma'(Z)\right|.$$

By Hoeffding's inequality,

$$\mathbb{P}_{\text{ho}}\left(\left|\frac{1}{n_{\text{ho}}}\sum_{i=1}^{n_{\text{ho}}} w^\top x_i \cdot y_i - \langle w_\star, \widehat{w}\rangle_\Sigma \mathbb{E}\sigma'(Z)\right| > \left|\langle w_\star, \widehat{w}\rangle_\Sigma \mathbb{E}\sigma'(Z)\right|\right)$$

$$\leq 2\exp\left(-\frac{cn_{\text{ho}}\left(\mathbb{E}\sigma'(Z)\right)^2 \langle w_\star, \widehat{w}\rangle_\Sigma^2}{\|\widehat{w}\|_\Sigma^2}\right) = 2\exp\left(-cn_{\text{ho}}\left(\cos\left(\theta_*\right)\cdot\mathbb{E}\sigma'(Z)\right)^2\right).$$

The theorem statement follows.

$\square$

# F Angular calibration for multi-index models

Let $x_{\text{new}} \sim \mathcal{N}(0, \Sigma) \subset \mathbb{R}^d$. Fix $K \geq 2$ and let $W_\star = [w_{\star 1}, \ldots, w_{\star K}] \in \mathbb{R}^{d \times K}$. Define

$$G := W_\star^\top x_{\text{new}} \in \mathbb{R}^K, \quad \pi(x_{\text{new}}) = g(G),$$

where $g$ is a generalized link (vector- or scalar-valued). This setup covers:

- Two-layer nets (frozen outer layer): $g(u) = \sum_k a_k \, \sigma(u_k)$
- Multi-class softmax: $g(u) = \text{softmax}(u)$
- Additive index model: $g(u) = \sum_k f_k(u_k)$
- Interaction index model: $g(u) = \sum_k f_k(u_k) + \sum_{k < \ell} h_{k\ell}(u_k, u_\ell)$

The following extends angular calibration to mutli-index models. Note that to apply the angular predictor (19), we must estimate cross-index angles $\langle w_{\star,k}, \widehat{w}_\ell \rangle_\Sigma$, similarly to the single-index case. Though no estimator is given in literature as far as we know, recent theory for multi-index models [82] suggests that analogues of the single-index angle estimators [8] are feasible. We leave derivation of these estimators to future works.

**Theorem F.1** (Angular calibration for multi-index models). *Let $\widehat{W} = [\widehat{w}_1, \ldots, \widehat{w}_K] \in \mathbb{R}^{d \times K}$ be any estimator. Set*

$$D := \text{diag}(\|\widehat{w}_1\|_\Sigma, \ldots, \|\widehat{w}_K\|_\Sigma), \quad S := D^{-1} \widehat{W}^\top x_{\text{new}} \in \mathbb{R}^K.$$

*Then we have $K \times K$ covariance blocks*

$$\text{Cov}(G) = W_\star^\top \Sigma W_\star, \quad R := \text{Cov}(S) = D^{-1} \widehat{W}^\top \Sigma \widehat{W} D^{-1}, \quad C := \text{Cov}(G, S) = W_\star^\top \Sigma \widehat{W} D^{-1}$$

*where*

$$R_{k\ell} = \frac{\langle \widehat{w}_k, \widehat{w}_\ell \rangle_\Sigma}{\|\widehat{w}_\ell\|_\Sigma \|\widehat{w}_k\|_\Sigma}, \qquad C_{k\ell} = \frac{\langle w_{\star k}, \widehat{w}_\ell \rangle_\Sigma}{\|\widehat{w}_\ell\|_\Sigma \|w_{\star k}\|_\Sigma}$$

*Then, assuming that $R$ is invertible, we may define*

$$M_\star := CR^{-1}, \qquad \Sigma_\star := \text{Cov}(G) - CR^{-1}C^\top$$

*and we have that $G \mid S \sim \mathcal{N}(M_\star S, \Sigma_\star)$. For any factor $L_\star$ with $L_\star L_\star^\top = \Sigma_\star$ and any $Z \sim \mathcal{N}(0, I_K)$ independent of everything, define the multi-index angular predictor*

$$\widehat{f}_{\text{ang}}(\widehat{W}^\top x_{\text{new}}) := \mathbb{E}_Z \left[ g(M_\star S + L_\star Z) \right]. \tag{19}$$

*Then for any $p \in \Delta^{K-1}$ and any $d, n \in \mathbb{N}_+$,*

$$p - \mathbb{E}\left[ \pi(x_{\text{new}}) \;\middle|\; \widehat{f}_{\text{ang}}(\widehat{W}^\top x_{\text{new}}) = p \right] = 0.$$

*Proof.* Let $x_{\text{new}} \sim \mathcal{N}(0, \Sigma)$ and define

$$G := W_\star^\top x_{\text{new}} \in \mathbb{R}^K, \qquad S := D^{-1} \widehat{W}^\top x_{\text{new}} \in \mathbb{R}^K,$$

with $D = \text{diag}(\|\widehat{w}_1\|_\Sigma, \ldots, \|\widehat{w}_K\|_\Sigma)$ and $\|u\|_\Sigma := \sqrt{u^\top \Sigma u}$, $\langle u, v \rangle_\Sigma := u^\top \Sigma v$. Set the $2K \times d$ linear map

$$T := \begin{bmatrix} W_\star^\top \\ D^{-1} \widehat{W}^\top \end{bmatrix}, \qquad Y := \begin{bmatrix} G \\ S \end{bmatrix} = T \, x_{\text{new}}.$$

Since $x_{\text{new}}$ is Gaussian and $Y$ is a linear transform, $Y$ is jointly Gaussian with mean 0 and covariance

$$\text{Cov}(Y) = T\Sigma T^\top = \begin{bmatrix} W_\star^\top \Sigma W_\star & W_\star^\top \Sigma \widehat{W} D^{-1} \\ D^{-1} \widehat{W}^\top \Sigma W_\star & D^{-1} \widehat{W}^\top \Sigma \widehat{W} D^{-1} \end{bmatrix}.$$

Thus,
$$\text{Cov}(G) = W_\star^\top \Sigma W_\star, \quad R := \text{Cov}(S) = D^{-1}\widehat{W}^\top \Sigma \widehat{W} D^{-1}, \quad C := \text{Cov}(G, S) = W_\star^\top \Sigma \widehat{W} D^{-1}.$$
In particular, for $k, \ell \in [K]$,
$$R_{k\ell} = \frac{\langle \widehat{w}_k, \widehat{w}_\ell \rangle_\Sigma}{\|\widehat{w}_k\|_\Sigma \|\widehat{w}_\ell\|_\Sigma}, \qquad C_{k\ell} = \frac{\langle w_{\star k}, \widehat{w}_\ell \rangle_\Sigma}{\|w_{\star k}\|_\Sigma \|\widehat{w}_\ell\|_\Sigma}.$$

Define the $K \times K$ matrices
$$M_\star := CR^{-1}, \qquad \Sigma_\star := \text{Cov}(G) - CR^{-1}C^\top.$$

Consider the linear residual
$$U := G - M_\star S = G - CR^{-1}S.$$
Because $Y$ is Gaussian, $U$ is Gaussian; further,
$$\text{Cov}(U, S) = \text{Cov}(G, S) - CR^{-1}\text{Cov}(S, S) = C - CR^{-1}R = 0,$$
so $U$ and $S$ are independent (uncorrelated jointly Gaussian vectors are independent). Moreover,
$$\text{Cov}(U) = \text{Cov}(G) - CR^{-1}C^\top = \Sigma_\star.$$
Hence we have the orthogonal decomposition
$$G = M_\star S + U, \qquad U \perp\!\!\!\perp S, \qquad U \sim \mathcal{N}(0, \Sigma_\star).$$
Equivalently, the conditional distribution is
$$G \mid S \sim \mathcal{N}(M_\star S, \ \Sigma_\star),$$
which is the standard multivariate normal conditioning formula via the Schur complement.

Finally, let $L_\star$ be any matrix satisfying $L_\star L_\star^\top = \Sigma_\star$ and let $Z \sim \mathcal{N}(0, I_K)$ independent of $(G, S)$. Then $U \overset{d}{=} L_\star Z$ and
$$G \mid S \overset{d}{=} M_\star S + L_\star Z.$$
Therefore, for any measurable $g$ (vector- or scalar-valued) with the requisite integrability,
$$\mathbb{E}[g(G) \mid S] = \mathbb{E}_Z[g(M_\star S + L_\star Z)],$$
which yields the stated multi-index angular predictor
$$\widehat{f}_{\text{ang}}(\widehat{W}^\top x_{\text{new}}) := \mathbb{E}_Z[g(M_\star S + L_\star Z)].$$

To conclude, set $X := \pi(x_{\text{new}}) = g(G) \in \Delta^{K-1}$ and
$$Y := \widehat{f}_{\text{ang}}(\widehat{W}^\top x_{\text{new}}) = \mathbb{E}[g(G) \mid S].$$
Then $Y$ is $\sigma(S)$-measurable and $Y = \mathbb{E}[X \mid S]$ almost surely. We claim that
$$\mathbb{E}[X \mid Y] = Y \qquad \text{a.s.}$$
Indeed, for any bounded measurable $\varphi : \Delta^{K-1} \to \mathbb{R}$,
$$\mathbb{E}[\varphi(Y)(X - Y)] = \mathbb{E}\big[\mathbb{E}[\varphi(Y)(X - Y) \mid S]\big] = \mathbb{E}[\varphi(Y)(\mathbb{E}[X \mid S] - Y)] = 0,$$
so $\mathbb{E}[X \mid Y] = Y$ (coordinate-wise) by the defining property of conditional expectation.

By the existence of regular conditional expectations, this implies that for $\mathbb{P}_Y$-almost every $p \in \Delta^{K-1}$,
$$\mathbb{E}\Big[\pi(x_{\text{new}}) \,\Big|\, \widehat{f}_{\text{ang}}(\widehat{W}^\top x_{\text{new}}) = p\Big] = \mathbb{E}[X \mid Y = p] = p,$$
i.e.,
$$p - \mathbb{E}\Big[\pi(x_{\text{new}}) \,\Big|\, \widehat{f}_{\text{ang}}(\widehat{W}^\top x_{\text{new}}) = p\Big] = 0.$$
This establishes exact calibration of the multi-index angular predictor. $\qquad\square$

# G Pseudocode

---

**Algorithm 1:** Angular Calibration

---

**Input** : Training data $\{(x_i, y_i)\}_{i=1}^{n}$; link $\sigma$; convex loss $\ell$ and penalty $g$;
covariance $\Sigma$ (known or estimated from unlabeled data);
holdout set $\{(x_i^{\mathrm{ho}}, y_i^{\mathrm{ho}})\}_{i=1}^{n_{\mathrm{ho}}}$ for sign estimation;

**Output :** Calibrated predictor $\hat{f}_{\mathrm{ang}}(x)$ that returns $\hat{p} \in [0, 1]$ for any new $x$.

**(A) Fit base linear model**

$\widehat{w} \leftarrow \arg\min_w \frac{1}{n} \sum_{i=1}^{n} \ell_{y_i}(w^\top x_i) + g(w)$

$\|\widehat{w}\|_\Sigma \leftarrow (\widehat{w}^\top \Sigma \, \widehat{w})^{1/2}$

**(B) Observable magnitude of $\langle w_\star, \widehat{w} \rangle_\Sigma$**

$\hat{\psi}_i \leftarrow -\ell'_{y_i}(x_i^\top \widehat{w}); \quad \hat{\psi} \leftarrow (\hat{\psi}_1, \ldots, \hat{\psi}_n)^\top$

$D \leftarrow \mathrm{diag}(\ell''_y(X\widehat{w})); \quad \hat{H} \leftarrow \left(X^\top D X + n \nabla^2 g(\widehat{w})\right)^{-1}$

$\hat{v} \leftarrow \frac{1}{n} \mathrm{Tr}(D - DX\hat{H}X^\top D); \quad \hat{\gamma} \leftarrow \mathrm{Tr}(X\hat{H}X^\top D); \quad \hat{r}^2 \leftarrow \|\hat{\psi}\|^2/n$

$\hat{a}_*^2 \leftarrow \dfrac{\left(\frac{\hat{v}}{n}\|X\widehat{w} - \hat{\gamma}\hat{\psi}\|^2 + \frac{1}{n}\hat{\psi}^\top X\widehat{w} - \hat{\gamma}\hat{r}^2\right)^2}{\frac{1}{n^2}\left\|\Sigma^{-\frac{1}{2}}X^\top\hat{\psi}\right\|^2 + \frac{2\hat{v}}{n}\hat{\psi}^\top X\widehat{w} + \frac{\hat{v}^2}{n}\|X\widehat{w} - \hat{\gamma}\hat{\psi}\|^2 - \frac{d}{n}\hat{r}^2}$  `// Est. of` $\langle w_\star, \widehat{w} \rangle_\Sigma^2$

**(C) Sign via holdout correlation**

$\widehat{\mathrm{sgn}} \leftarrow \mathrm{sign}\left(\sum_{i=1}^{n_{\mathrm{ho}}}(\widehat{w}^\top x_i^{\mathrm{ho}}) \, y_i^{\mathrm{ho}}\right)$

**(D) Angle & interpolation weights**

$c \leftarrow \dfrac{\widehat{\mathrm{sgn}} \sqrt{\hat{a}_*^2}}{\|\widehat{w}\|_\Sigma}; \quad c \leftarrow \min\{1, \max\{-1, c\}\}$      `// numerical clip`

$\hat{\theta} \leftarrow \arccos(c); \quad \alpha \leftarrow \cos\hat{\theta} = c; \quad \beta \leftarrow \sin\hat{\theta} = \sqrt{1 - \alpha^2}$

**(E) Define calibrated predictor $\hat{f}_{\mathrm{ang}}$**

*For any new $x$:*

$u \leftarrow \widehat{w}^\top x; \quad s \leftarrow u/\|\widehat{w}\|_\Sigma$

draw $z_1, \ldots, z_M \overset{iid}{\sim} \mathcal{N}(0, 1)$ and set $\hat{f}_{\mathrm{ang}}(x) \approx \frac{1}{M}\sum_{j=1}^{M} \sigma(\alpha s + \beta z_j)$

**return** $\hat{f}_{\mathrm{ang}}(x)$

---

# H  Additional plots

## H.1  Universality

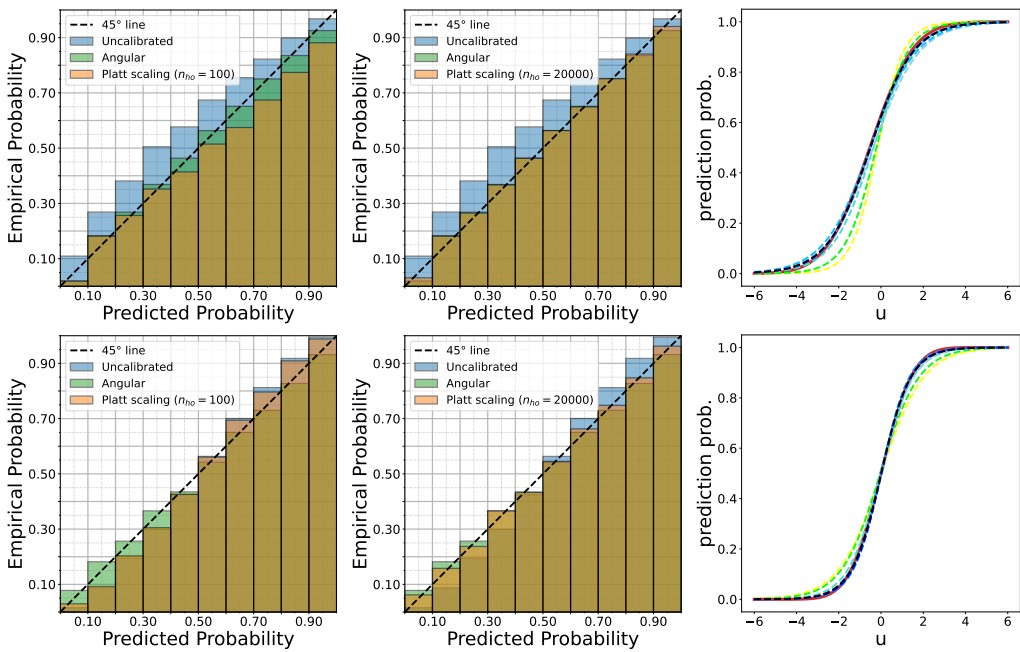

Figure 3: Reproduce Figure 1 (in the third column) and 2 (in first two columns) for Rademacher entries. Upper Row: rerun simulations in Section 7 but with subGaussian designs $W\Sigma^{1/2}$ where $W_{ij}$ are sampled iid from Rademacher distribution, taking values $+1, -1$ with equal probability. Bottom Row: we replace the sigmoid link function in Section 7 with a clipped relu link function $\sigma(x) = \mathrm{clip}(3x + 0.5)$ where $\mathrm{clip}(x) = x, \forall x \in [0, 1]$, $\mathrm{clip}(x) = 0, \forall x < 0$ and $\mathrm{clip}(x) = 1, \forall x > 1$.

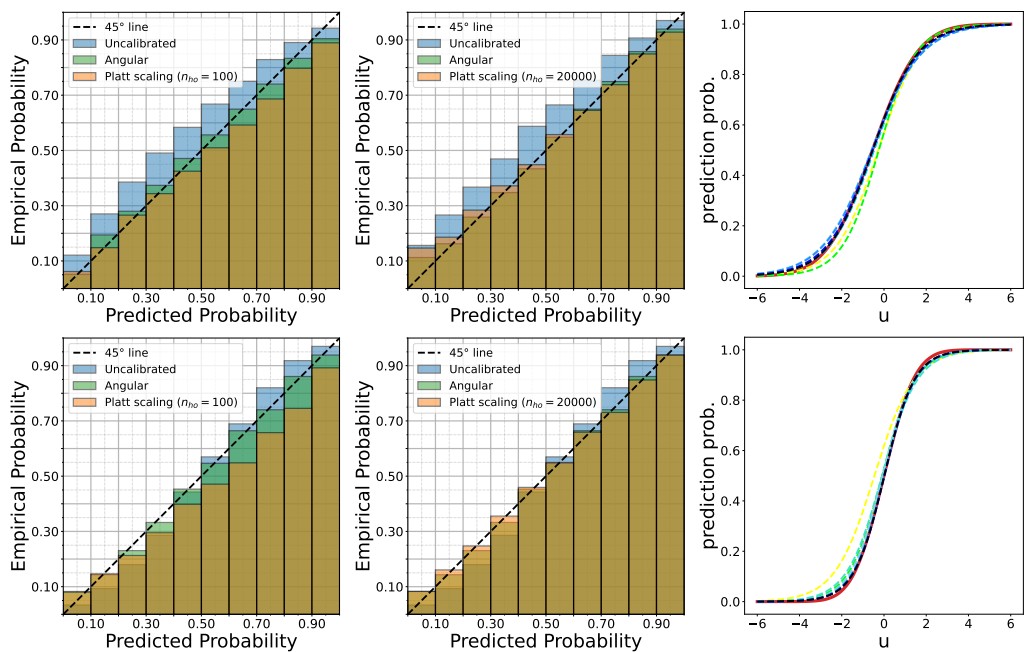

Figure 4: Reproduce Figure 1 (the third column) and 2 (first two columns) for uniform entries. Upper Row: rerun simulations in Section 7 but with non-Gaussian designs $W\Sigma^{1/2}$ where $W_{ij}$ are sampled iid from uniform distribution, taking values in interval $[-\sqrt{12}/2, \sqrt{12}/2]$ uniformly at random. Bottom Row: we replace the sigmoid link function in Section 7 with a clipped relu link function $\sigma(x) = \text{clip}(3x + 0.5)$ where $\text{clip}(x) = x, \forall x \in [0,1]$, $\text{clip}(x) = 0, \forall x < 0$ and $\text{clip}(x) = 1, \forall x > 1$.

## H.2  UCI benchmarks and pretrained embeddings

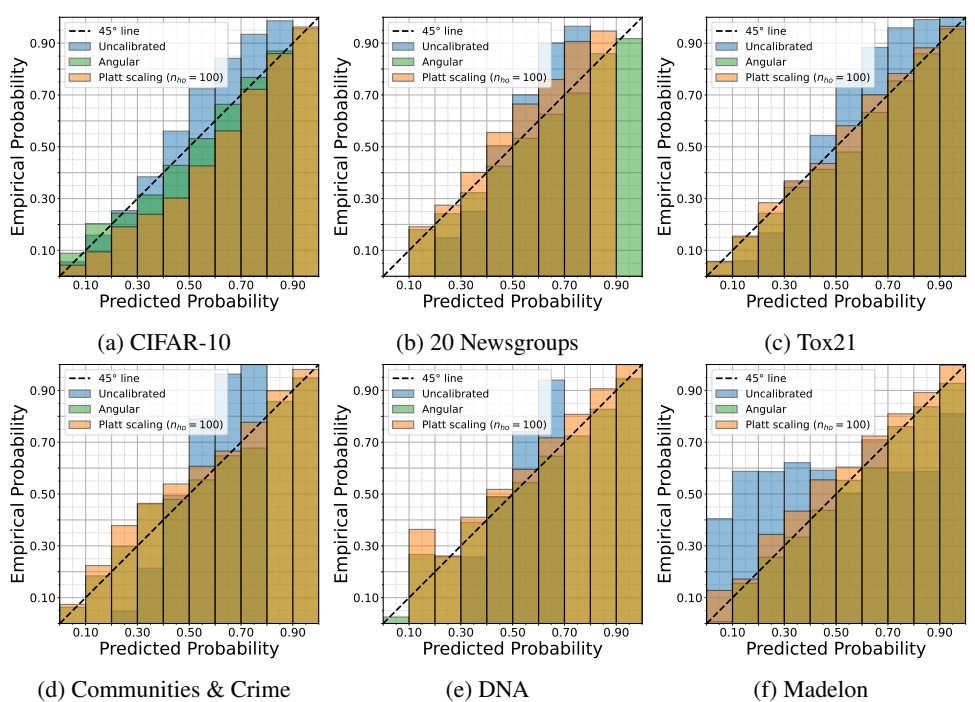

(a) CIFAR-10  (b) 20 Newsgroups  (c) Tox21

(d) Communities & Crime  (e) DNA  (f) Madelon

# I  Simulation code and reproduction

The simulation code Angular_calibration.ipynb, Semi_experiments.ipynb included as supplementary material. The experiment can be run on modern personal computers without needing special computing hardware. Detailed instructions is provided in the simulation code.

