# OpenReview forum: "Optimal and Provable Calibration in High-Dimensional Binary Classification: Angular Calibration and Platt Scaling"
_NeurIPS.cc/2025/Conference — NeurIPS 2025 spotlight_

### Official Review · Reviewer_Zw5n · 2025-06-21

**Clarity:** 4
**Significance:** 3
**Originality:** 3
**Rating:** 5
**Confidence:** 3

**Summary:**

The paper studies calibration in the setting that the target labels are sampled from a single-index link-regression model under gaussian inputs. This work gives a clean method for transforming the predictions from a linear model such that the predictions converge to the optimal transformation with respect to Bregman divergence to the underlying distribution of the labels.  They demonstrate theoretically and empirically that Platt scaling converges to their angular calibration method with infinite holdout samples. The authors provide experimental validation for their algorithm and theoretical results.

**Questions:**

1.	I am not sure I understand the optimality/consistency result. As the samples go to infinity, wouldn’t it be the case that any estimation method that has small error to the true predictor be perfectly calibrated given access to the link function as assumed for your angular predictor? Or is the consistency result saying that even if you have large error to the true w*, you will achieve optimal calibration provided n,d are infinite?
2.	Your calibration method will also requires data to estimate the angular distance of \hat{w} to w*. Is the advantage then that you don’t require holdout data to compute this distance, whereas Platt scaling does?
3.	How are calibration guarantees affected when the inputs to your calibration method is not exactly linear but instead the outputs of some nonlinear model? I am imagining transforming predictions with logistic regression as in Platt Calibration.

**Ethical Concerns:**

["NO or VERY MINOR ethics concerns only"]

**Final Justification:**

As stated above, the paper introduces a clean optimal calibration method (assuming access to the known linear classifier). This method justifies Platt scaling which is a widely used method for calibration. I increased my score as they demonstrated experiments for their method on more realistic settings (logits of neural networks) and theoretical results for multi-index models.

**Limitations:**

Yes.

**Paper Formatting Concerns:**

None.

**Quality:**

4

**Strengths And Weaknesses:**

If my questions are addressed adequately, I will increase my score.

Strengths:\
-	The problem tackled here is fundamental.\
-	Their approach is quite clean.\
-	Their approach justifies Platt Scaling which is a very common method for calibrating predictors of all types.

Weaknesses:\
-	The calibration method proposed requires knowledge of the link function, which seems like a strong assumption, especially when you are applying this predictions to calibrate the outputs of some nonlinear predictor. \
-	Their method requires that the true function is linear with some link function that angular calibration method has access to. Their result might be made much stronger if the authors could address if and how their results extend to the nonlinear case.\
-	Experiments are limited to the realizable case where the target is indeed some logistic regression function, and not in the standard calibration setting where your inputs are themselves predictions from some nonlinear method.

---

> ### Author Rebuttal · Authors · 2025-07-27
>
> We are grateful for the reviewer’s thoughtful review, and ask that the reviewer consider raising their score if they found our response satisfactory.
>
> ## R1. Weakness
>
> 1. We agree with the reviewer’s comment regarding the limitations of angular calibration—it indeed assumes knowledge of the true link function. However, recent developments in the literature \[4] suggest that it is possible to consistently estimate the true link function in the proportional regime. One could, in principle, apply such a method to first estimate the link function and then use it in angular calibration. We also present a simulation in **Section R5**, showing that the expected calibration error of angular calibration increases approximately linearly with the estimation error of the true link function.
>
> 2. Our setting is the simplest analytically tractable regime for studying calibration in high dimensions. Even here, no existing method is provably calibrated. Additionally, our theory extends to multi-index models, covering more complex structures such as two-layer neural nets (with frozen outer layer), additive, and interaction models. See **Section R3** below.
>
> 3. While derived for linear predictors, angular calibration can be applied to final-layer logits of deep nets—a standard practice. See **Section R4** for experiments using angular calibration (as well as isotonic regression and Platt scaling) on logits from **ResNet34**, **MiniLM**, and **ChemBERTa** over **CIFAR10**, **20newsgroups**, and **Tox21**. There, we also include additional experiments on UCI benchmark datasets: **Communities & Crime**, **Splice-junction**, **Madelon**, **ISOLET**.
>
>
> ## R2. Questions
>
> 1. The latter is correct. In the proportional regime, the estimated signal $\hat{w}$ does not converge to the true $w^\star$ as both $n$ and $d$ grow proportionally. In fact, under our model assumptions, $\langle \hat{w}, w^\star \rangle / (\|\hat{w}\|\|w^\star \|)$ converges to a constant less than 1. Nevertheless, angular calibration achieves asymptotic optimality in the sense of minimizing among all calibrated predictor of the form $F(x^\top_{new} \hat{w})$ expected Bregman divergence
>
> 2. Yes, exactly. Angular calibration does not require a validation set.
>
> 3. Yes, one may apply angular calibration to the last layer logits output of modern deep learning models. One would essentially treat these logits as input data. See **Section R4** for our results. While we found that estimator (11) for $\langle w^\star, \hat w \rangle$ is unstable on real data, angular calibration remains robust when this angle is known accurately.
>
> ## R3 Multi-index model
>
> **(Theorem R1. Angular calibration for multi-index models).**  Let $\widehat W=[\widehat w_1,\ldots,\widehat w_K]\in\mathbb R^{d\times K}$ be any estimator. Set
> $$
> D:=\operatorname{diag}(\rVert \widehat w_1 \rVert_\Sigma, \ldots, \rVert \widehat w_K \rVert_\Sigma), \quad
> S := D^{-1}\widehat W^\top x_{\mathrm{new}} \in \mathbb{R}^K.
> $$
> Then we have $K\times K$ covariance blocks
> $$
> \operatorname{Cov}(G)=W_\star^\top\Sigma W_\star,\quad
> R:=\operatorname{Cov}(S)=D^{-1}\widehat W^\top\Sigma \widehat W D^{-1},\quad
> C:=\operatorname{Cov}(G,S)=W_\star^\top\Sigma \widehat W D^{-1}
> $$
> where
>
> $$R_{k\ell}=\frac{\langle \widehat w_k,\widehat{w}_\ell\rangle _{\Sigma}}{\lVert\widehat{w} _{\ell}\rVert _{\Sigma} \lVert\widehat{w}_k \rVert _{\Sigma}}$$
>
> $$
> C_{k\ell}=\frac{\langle w_{\star k},\widehat{w}_\ell \rangle _\Sigma}{\lVert\widehat{w} _{\ell}\rVert _{\Sigma} \lVert w _{\star k} \rVert _{\Sigma}}
> $$
>
>
> Then,
>
> $$
> \ M_\star:=C R^{-1},\qquad
> \Sigma_\star:=\operatorname{Cov}(G)-C R^{-1}C^\top\
> $$
>
> satisfy $G\mid S\sim\mathcal N(M_\star S,\Sigma_\star)$. For any factor $L_\star$ with $L_\star L_\star^\top=\Sigma_\star$ and any $Z\sim\mathcal N(0,I_K)$ independent of everything, define the multi-index angular predictor
>
> $$
> \ \widehat f_{\mathrm{ang}}\big(\widehat W^\top x_{\mathrm{new}}\big)
> :=\mathbb E_{Z}\left[g\big(M_\star S+L_\star Z\big)\right]. \
> $$
>
> Then for any $p\in\Delta^{K-1}$ and any $d,n\in\mathbb N_+$,
>
> $$
> p-\mathbb E\left[\pi(x_{\mathrm{new}})\ \middle|\ \widehat f_{\mathrm{ang}}(\widehat W^\top x_{\mathrm{new}})=p\right]=0.
> $$
>
> To apply the above, we must estimate \$\langle \widehat w_k,\widehat w_\ell\rangle _\Sigma\$, \$\forall k,\ell\in\[K]\$, as in the single-index case. Though no estimator is given in literature as far as we know, recent theory for multi-index models \[1] suggests that analogues of the single-index angle estimators \[2] are feasible.
>
> ## R4. Semi-real experiments
>
> For Tables 1–2, we subsample each dataset into training (\$n\_{\text{train}}\$), covariance estimation (\$n\_{\text{cov}}\$), and test (\$n\_{\text{test}}\$) sets. We set \$n\_{\text{cov}} \gg n\_{\text{train}}\$ to reflect abundant unlabeled data, and report ECE for \$n\_{\text{ho}} = 100\$ and \$500\$. We report calibration quality using Expected Calibration Error (ECE), which measures the average gap between predicted confidence and actual accuracy across confidence levels, quantifying how well a model's probabilities are calibrated \[5].
>
> We found that Eq. (11)’s plug-in estimator for \$\langle w^\star, \hat{w} \rangle\$ is unstable on real datasets. This is a known issue for estimators like (11) that are based on Wigner-type random-matrix-theoretic assumptions. Modifying these estimator are an active research area \[3]. To isolate calibration effects, we simulate \$w^\star\$ and labels, using the *true* angle.
>
> To extend calibration to nonlinear models, we apply it to the logits of pretrained models:
>
> * **ResNet-34 → CIFAR-10**: Use 512-dim penultimate layer of ResNet-34 (ImageNet pretrained)
> * **MiniLM → 20 Newsgroups**: Mean-pool 384-dim “all‑MiniLM‑L6‑v2” embeddings
> * **ChemBERTa → Tox21**: Mean-pool 768-dim hidden states from “seyonec/ChemBERTa‑zinc‑base‑v1”
>
> \$n\_{\text{cov}} = 30000\$, \$3000\$, \$3000\$; \$n\_{\text{test}} = 10000\$, \$1000\$, \$500\$
>
> *Table 1.* ECE for pretrained DL models:
>
> | Model–Dataset       | \$n \times d\$ | Uncal  | Angular | Platt 100 | Iso 100 | Platt 500 | Iso 500 |
> | ------------------- | -------------- | ------ | ------- | --------- | ------- | --------- | ------- |
> | ResNet34–CIFAR10    | 300×512        | 0.1236 | 0.0199  | 0.0561    | 0.0484  | 0.0259    | 0.0298  |
> | MiniLM–20newsgroups | 800×384        | 0.1392 | 0.0249  | 0.0931    | 0.1107  | 0.0679    | 0.0813  |
> | ChemBERTa–Tox21     | 800×768        | 0.1389 | 0.0132  | 0.0236    | 0.0497  | 0.0175    | 0.0293  |
>
> For UCI datasets:
>
> * Communities & Crime (200×100), Splice-junction (100×180), Madelon (300×500), ISOLET (500×617)
> * \$n\_{\text{cov}} = 700\$, \$2000\$, \$900\$, \$5750\$; \$n\_{\text{test}} = 593\$, \$900\$, \$900\$, \$1000\$
>
> *Table 2.* ECE on UCI datasets:
>
> | UCI Dataset              | \$n \times d\$ | Uncal  | Angular | Platt 100 | Iso 100 | Platt 500 | Iso 500 |
> | ------------------------ | -------------- | ------ | ------- | --------- | ------- | --------- | ------- |
> | Communities and Crime    | 200×100        | 0.1700 | 0.0296  | 0.0620    | 0.0661  | 0.0262    | 0.0436  |
> | Splice-junction Gene Seq | 100×180        | 0.1262 | 0.0208  | 0.0568    | 0.1041  | 0.0556    | 0.0879  |
> | Madelon                  | 300×500        | 0.1473 | 0.0267  | 0.0721    | 0.0719  | 0.0299    | 0.0624  |
> | ISOLET                   | 500×617        | 0.1256 | 0.0221  | 0.0438    | 0.0761  | 0.0219    | 0.0137  |
>
> ## R5. Effect of inaccurate link function estimation
>
> We model the the estimated link function as follows:
>
> $$\hat \sigma (u)=(1+\varsigma \cdot \epsilon)\cdot \sigma(u), \qquad \epsilon \sim N(0,1)$$
>
> where $\varsigma \cdot \epsilon$ models estimation noise. We run a sequence of experiments at different $\varsigma$, each experiment repeated 15 times to produce mean ± std of Expected Calibration Error (ECE) of angular calibration. Otherwise, the setting is the same as Section 7 of the paper.
>
> Using the data below, we may fit a power law relationship of how ECE relates to the noise size $\varsigma$:
>
> $$\mathrm{ECE} \approx 0.183 \cdot \varsigma^{0.981}$$
>
> This seems to suggest that the expected calibration error grows linearly with estimation error of the link function.
>
>
> **Table.** Expected Calibration Error (ECE) vs. Estimation Noise $\varsigma$
>
> | Estimation noise $\varsigma$ | 0        | 0.0707           | 0.1000           | 0.1414           | 0.2000           | 0.2828           | 0.4000           | 0.5657           | 0.8000           |
> |------------------------------|----------|------------------|------------------|------------------|------------------|------------------|------------------|------------------|------------------|
> | ECE                          | 0.0077 ± 0.0 | 0.0081 ± 0.0008 | 0.0108 ± 0.0009 | 0.0182 ± 0.0010 | 0.0321 ± 0.0011 | 0.0560 ± 0.0013 | 0.0860 ± 0.0016 | 0.1164 ± 0.0018 | 0.1358 ± 0.0024 |
>
>
>
> ## References
> \[1] Troiani et al., 2024. *Fundamental computational limits of weak learnability in high-dimensional multi-index models*. arXiv:2405.15480
>
> \[2] Bellec, 2022. *Observable adjustments in single-index models*. arXiv:2204.06990
>
> \[3] Li & Sur, 2023. *Spectrum-aware debiasing*. arXiv:2309.07810
>
> \[4] Sawaya et al. *High-Dimensional Single-Index Models: Link Estimation and Marginal Inference*, arXiv:2404.17812
>
> \[5] Pavlovic. *Understanding Model Calibration - A gentle introduction and visual exploration of calibration and the expected calibration error (ECE)* arXiv:2501.19047v2

---

> > ### Author Response · Authors · 2025-08-03
> >
> > We would like to follow up briefly to check whether the reviewer found our response satisfactory and if there are any remaining questions. If so, we would be grateful if the reviewer would consider raising their score.

---

> > > ### Comment · Reviewer_Zw5n · 2025-08-05
> > > **Reviewer response**
> > >
> > > Thank you for your reply. Your rebuttal addresses my concerns. I will raise my score 1 point.

---

> > > > ### Author Response · Authors · 2025-08-06
> > > >
> > > > We thank the reviewer for their careful reading of our work and thoughtful feedback!

---

### Official Review · Reviewer_jieG · 2025-06-23

**Clarity:** 4
**Significance:** 4
**Originality:** 4
**Rating:** 6
**Confidence:** 5

**Summary:**

This works concerns the problem of estimating class probability in binary classification in the regime where both # of features and sample diverge to infinity.
The focus is on an angle-with-the-ground-truth-based estimator (or as the authors call it angular estimator simply).

**Questions:**

question:
- in theorem 3.2, is monotonicity of sigma not needed? it is quite surprising to me that continuity is all that is needed.
- for the convex penalty g (from line 82), can it be arbitrary as long as it is convex?
- what is the connection to "The Isotron Algorithm: High-Dimensional Isotonic Regression" by Kalai and Sastry? Since isotonic regression also targets the probability calibration, might be good to talk about similarity/differences of your work and Kalai and Sastry.

suggestion:
- line 60 maybe qualify your usage of the word "interpolating" since its usage is a bit overloaded in the literature. perhaps a footnote or a parenthetical remark that interpolation in this context means a convex combination
- it will be helpful to have an algorithm-style write up of the angular estimator. right now, it takes some work to chase it down. would be good to make it a bit more explicitly spelled out.

**Ethical Concerns:**

["NO or VERY MINOR ethics concerns only"]

**Final Justification:**

The authors provided a thorough rebuttal and additional discussion for the multiclass case which is exemplary.

**Limitations:**

Yes

**Quality:**

4

**Strengths And Weaknesses:**

Strength:
The result is quite beautiful and gives a surprising (but makes sense in hindsight) intuition about probability calibration on Line 128 - 129: "when the estimator is poorly aligned, the estimation must be tempered with noise"  (quote is this reviewer's paraphrasing).
The authors provide theory for platt scaling by showing (under certain parametric conditions) that platt scaling can be seen as an instance of the angle-based estimator.

Weakness:
The theoretical results are for the gaussian features setting, which is not as broad as one would hope. But the gaussianity assumption is common in the random matrix-theoretic literature, and the phenomenon discovered often experimentally can be observed even without the Gaussianity assumption.
Literature review is pretty thorough, but I feel that "The Isotron Algorithm: High-Dimensional Isotonic Regression" by Kalai and Sastry is quite related.

---

> ### Author Rebuttal · Authors · 2025-07-30
>
> We are grateful for the reviewer’s favorable review and encouraging comments.
>
> To answer the reviewer's questions:
>
> 1. Monotonicity of the link is not needed for Theorem 3.2. The result follows from the general, exact‑calibration statement in Theorem A.1 together with consistency of the angle estimator, using the tower property and the conditional law of jointly Gaussian variables. Continuity of the link is required only to pass the limit via the continuous‑mapping argument when the angle estimator in (11) converges.
>
> 2. Not quite. The theory in \[1] ensures consistency of the angle estimator only when (i) $g$ is strongly convex (Theorem 4.4 of \[1]), or (ii) $g=0$ and $\hat w$ exists (Theorem 5.2 of \[1]). Outside these cases, the guarantee does not apply.
>
> 3. We thank the reviewer for the reference and will add the citation. Kalai–Sastry’s Isotron alternates perceptron‑style updates with isotonic regression to learn a single‑index model; like our work, it exploits the single‑index structure by operating on the one‑dimensional score $w^\top x$ and fitting a monotone map on that score. However, while its analysis yields dimension‑free bounds on cumulative squared error and PAC‑style learnability for Lipschitz SIMs, it provides neither population reliability guarantees nor optimality results in the proportional high‑dimensional regime ($n,d\to\infty$ with $n/d$ fixed). By contrast, we study post‑hoc calibration of a fixed linear predictor in this regime via angular calibration, proving asymptotic calibration and Bregman‑optimality, and showing that Platt scaling converges to the same limit under suitable conditions.
>
> We thank the reviewer for suggestions on qualifying the "interpolation" and including an algorithm write-up. We will include them in later version of the draft.
>
> \[1] Bellec, 2022. Observable adjustments in single-index models. arXiv:2204.06990

---

> > ### Comment · Reviewer_jieG · 2025-08-05
> >
> > Thanks to the authors for the rebuttal. I maintain my original positive score.
> >
> > I also echo the other reviewer regarding extension to multiclass: it'd be useful to have a slightly more extended discussion about this, e.g., difficulty of the extension, what the authors think might be promising, etc. Has there been any published work on multiclass Platt scaling (even just introducing it)? Would be beneficial to the community to have a remark on this matter one way or another.

---

> ### Author Response · Authors · 2025-08-06
>
> We thank the reviewer for their positive feedback and for the questions on the multiclass extension.
>
> **(Multi-class angular calibration and challenges)** As discussed in Section R3 in our response to reviewer thu8, a $K$-class classification setting may be modeled as follows. Fix \$K\ge2\$ and let \$W\_\star=\[w\_{\star 1},\ldots,w\_{\star K}]\in\mathbb R^{d\times K}\$. Define the true logits and model output:
> $$
> G:=W_\star^\top x_{\mathrm{new}}\in\mathbb R^K,\quad \pi(x_{\mathrm{new}})=g(G)
> $$
> where \$g\$ is a generalized link (vector- or scalar-valued). Given an estimator $\widehat W=[\widehat w_1,\ldots,\widehat w_K] $ of $W_\star$ , angular predictor presented our paper may be extended to the multi-class version $\ \widehat f_{\mathrm{ang}}\big(\widehat W^\top x_{\mathrm{new}}\big)
> :=\mathbb E_{Z}\left[g\big(M_\star S+L_\star Z\big)\right]$ where the matrix quantity $M_\star$ and $L_\star$ depends on cross-class angles $\langle w_{\star k},\widehat{w}_\ell \rangle _\Sigma, \ell, k \in [K]$.
>
> The derivation is straightforward: we generalize identity (15) in Theorem A.1 by replacing the scalar conditioning $\left[w_{\star}^{\top} x_{\text {new }} \mid x_{\text {new }}^{\top} \widehat{w}\right]$ with its multivariate analogue, conditioning the true multiclass logits $G:=W_\star^\top x_{\mathrm{new}}$ on the normalized score vector $S := D^{-1}\widehat W^\top x_{\mathrm{new}}$ where $D:=\operatorname{diag}(\rVert \widehat w_1 \rVert_\Sigma, \ldots, \rVert \widehat w_K \rVert_\Sigma)$.
>
> We will incorporate Section R3 for reviewer thu8 and supply a detailed proof in the later versions of the paper.
>
> The main remaining challenge is to consistently estimate the unobserved cross-class angles $\langle w_{\star k}, \widehat w_\ell  \rangle_\Sigma$ for $k,\ell\in[K]$, which essentially boils down to extending \[1] from the single-index to the multi-index setting. While we view this as a promising direction—even perhaps incremental—it still likely requires involved technical analysis as in \[1].
>
> **(Existing multi-class Platt scaling and related methods)** Regarding multiclass Platt scaling: yes, there is prior work. A standard approach fits binary Platt scalers and then combines them into multiclass probabilities via one-vs-rest normalization or pairwise coupling (e.g., \[2,3,4]). More recently, temperature scaling (one parameter on logits) and its multiclass generalizations—vector scaling and matrix scaling—are widely used on deep-net logits \[5,6]. There are also explicitly multiclass post-hoc maps such as Dirichlet calibration \[7]. We will add a short remark situating our multiclass construction relative to these methods.
>
> **(Future directions)** Promising directions include: (i) developing consistent estimators of cross-class angles in the proportional regime; (ii) establishing Bregman optimality for the multi-index angular predictor and connecting it to vector/matrix/Dirichlet scaling (in parallel to Sections 4–5 of our paper); and (iii) formalizing universality and deriving finite-sample guarantees. More open-ended avenues are: (a) exploring possibility of extending the idea to predictive inference \[8]; and (b) empirically studying, or perhaps adapting, angular calibration for specific ML subdomains \[9-11].
>
>
> **References**
>
> \[1] Bellec, 2022. Observable adjustments in single-index models. arXiv:2204.06990
>
> \[2] Zadrozny, B. & Elkan, C., 2002. Transforming classifier scores into accurate multiclass probability estimates. *Proceedings of the Eighth ACM SIGKDD International Conference on Knowledge Discovery and Data Mining (KDD ’02)*, pp. 694–699.
>
> \[3] Hastie, T. & Tibshirani, R., 1998. Classification by pairwise coupling. *The Annals of Statistics*, 26(2), pp. 451–471.
>
> \[4] Wu, T.-F., Lin, C.-J. & Weng, R.C., 2004. Probability estimates for multi-class classification by pairwise coupling. *Journal of Machine Learning Research*, 5, pp. 975–1005.
>
> \[5] Guo, C., Pleiss, G., Sun, Y. & Weinberger, K.Q., 2017. On calibration of modern neural networks. *Proceedings of the 34th International Conference on Machine Learning (ICML 2017)*, PMLR 70, pp. 1321–1330.
>
> \[6] Nixon, J., Dusenberry, M.W., Jerfel, G., Liu, J., Zhang, L. & Tran, D., 2019. Measuring calibration in deep learning. arXiv preprint arXiv:1904.01685.
>
> \[7] Kull, M., Perelló-Nieto, M., Kängsepp, M., Silva Filho, T.M., Song, H. & Flach, P., 2019. Beyond temperature scaling: Obtaining well-calibrated multiclass probabilities with Dirichlet calibration. *NeurIPS 2019*.
>
> \[8] Distribution-free binary classification: prediction sets, confidence intervals and calibration, Neurips, 2020
>
> \[9] QA-Calibration of Language Model Confidence Scores, Intl. Conf. on Learning Representations (ICLR), 2025
>
> \[10] Faster online calibration without randomization, Conf. on Learning Theory (COLT), 2022
>
> \[11] Küppers, F., Kronenberger, J., Shantia, A. & Haselhoff, A., 2020. Multivariate confidence calibration for object detection. arXiv preprint arXiv:2004.13546.

---

### Official Review · Reviewer_thu8 · 2025-06-29

**Clarity:** 3
**Significance:** 3
**Originality:** 4
**Rating:** 5
**Confidence:** 4

**Summary:**

This paper addresses the problem of calibration in binary classification, focusing on classifiers that are linear with respect to input features. It investigates the question: What is the best calibrated predictor achievable via a post-hoc parametric calibration function that maps the original prediction to a calibrated probability? To answer this, the paper introduces the notion of angular calibration, where it can be shown that calibration with this notion will achieve both calibration and Bregman-optimality simultaneously. Theoretical analysis and simulations further show that, given sufficient data, Platt scaling converges to the angular calibration solution.

**Questions:**

Questions:
1. Extension to Multiclass Setting
Are there any significant bottlenecks or theoretical challenges that make it difficult to extend the proposed method to the multiclass setting? Do you think algorithm for computing angular predictor will change significantly or very difficult to derive in this setting? If so, it would be valuable to include a discussion of these challenges in the paper.
2. Calibration Metrics
While reliability diagrams are useful, they can be difficult to interpret when curves overlap. It would be helpful to also report a summary metric such as Expected Calibration Error (ECE). This could be included in the figure caption or legend to save space, and would make it easier to compare methods quantitatively.
3. Dimensionality of Simulated Data
I may have missed it, but it would be helpful to clarify the dimensionality of the simulated data used in the experiments. Given the paper's focus on high-dimensional settings, this is an important detail.
4. Effect of Varying Dimensions
It could be insightful to explore how calibration performance (e.g., using ECE) changes with different feature dimensions—such as low, moderate, and very high dimensions. This would provide a deeper understanding of how the proposed method behaves under varying regimes.
5. Use of Benchmark Datasets
The assumption of Gaussian features is indeed quite restrictive, and I appreciate that this limitation is acknowledged in the paper. I also found Figures 3 and 4 helpful in demonstrating that the proposed method performs well even under non-Gaussian settings. That said, I believe the use of a standard benchmark dataset such as those from the UCI repository could provide additional insights and help better motivate the practical utility of the proposed method. This would also make it easier to compare with other methods and assess generalizability in more realistic scenarios. What are your thoughts on this?

Suggestion:
I suggest including a summary of the proposed algorithm in a LaTeX algorithm environment. This would help readers quickly grasp the procedure for computing the angular predictor and enhance reproducibility.

**Ethical Concerns:**

["NO or VERY MINOR ethics concerns only"]

**Final Justification:**

After reading the rebuttal of the authors, I found that the reply was satisfactory. My all three concerns were addressed, which are the concern of the difficulty of extension to multiclass, extension to non-linear models, and the evaluation beyond a synthetic dataset. I increased the score to 5.

**Limitations:**

yes

**Quality:**

3

**Strengths And Weaknesses:**

Strengths
1. The paper is well-written and easy to follow.
2. The theoretical results are clearly presented and, to the best of my knowledge, novel and innovative.
3. The proposed algorithm is directly derived from the theoretical insights and is both simple and effective.

Weaknesses
1. The theoretical analysis is limited to binary classification.
2. The assumption of a linear predictor restricts the applicability of the results to modern machine learning models, which often rely on more expressive architectures.
3. The empirical evaluation is limited to toy datasets, raising concerns about practical relevance and generalizability.

Overall Evaluation
I thoroughly enjoyed reading this paper and found the theoretical contributions fascinating. However, the overall significance may be somewhat limited due to the restrictive assumptions and lack of empirical validation on real-world datasets. My current assessment, prior to the rebuttal, is borderline accept.

---

> ### Author Rebuttal · Authors · 2025-07-27
>
> We are grateful for the reviewer’s thoughtful review, and ask that the reviewer consider raising their score if they found our response satisfactory.
>
> ## R1. Weakness
>
> 1. Theorems 1 and 8 naturally extend to **multi-index models**, including multi-class classification \[1]; see **Theorem R1**.
> 2. Our setting is the simplest analytically tractable regime for studying calibration under overparametrization. Even here, no existing method is provably calibrated. Additionally:
>
>    * (i) Our theory extends to multi-index models, which includes multi-class classification and covers more complex structures such as two-layer neural nets (with frozen outer layer), additive, and interaction index models (see **Section R3**).
>    * (ii) While derived for linear predictors, angular calibration can be applied to final-layer logits of deep nets—a standard practice. See **Section R4** for experiments using angular calibration (as well as isotonic regression and Platt scaling) on logits from **ResNet34**, **MiniLM**, and **ChemBERTa** over **CIFAR10**, **20newsgroups**, and **Tox21**.
> 3. See **Sections R4** for additional experiments (UCI datasets: **Communities & Crime**, **Splice-junction**, **Madelon**, **ISOLET**) .
>
> ## R2. Questions
>
> 1. Yes, angular calibration extends to multi-class via the multi-index framework; see **Section R3**.
> 2. Thank you for the suggestion. We have implemented this and will include in camera ready version.
> 3. Section 7 specifies \$n=1000\$, \$d=2000\$. Additional experiments varying \$n, d\$ are in **Section R5**.
> 4. See **Tables 3–4** in **Section R5**, comparing angular, Platt, and isotonic calibration across \$n, d\$. Angular calibration performs best when \$d/n \in \[0.05, 8]\$ and deteriorates outside this range.
> 5. See **Section R4** for UCI experiments. While estimator (11) for \$\langle w^\star, \hat w \rangle\$ is unstable on real data, angular calibration remains robust when this angle is known accurately.
>
> We will include the proposed algorithm in a LaTeX algorithm environment in camera-ready version.
>
> ## R3. Multi-index models
>
> Let \$x\_{\mathrm{new}}\sim\mathcal N(0,\Sigma)\$. Fix \$K\ge2\$ and let \$W\_\star=\[w\_{\star 1},\ldots,w\_{\star K}]\in\mathbb R^{d\times K}\$. Define the true logits and model output:
>
> $$
> G:=W_\star^\top x_{\mathrm{new}}\in\mathbb R^K,\quad \pi(x_{\mathrm{new}})=g(G)
> $$
>
> where \$g\$ is a generalized link (vector- or scalar-valued). This setup covers:
>
> * Two-layer neural nets with frozen outer layer: \$g(u)=\sum\_k a\_k \sigma(u\_k)\$
> * Multi-class softmax: \$g(u)=\mathrm{softmax}(u)\$
> * Additive model: \$g(u)=\sum\_k f\_k(u\_k)\$
> * Interaction model: \$g(u)=\sum\_k f\_k(u\_k)+\sum\_{k<\ell} h\_{k\ell}(u\_k,u\_\ell)\$
>
> **(Theorem R1. Angular calibration for multi-index models).**  Let $\widehat W=[\widehat w_1,\ldots,\widehat w_K]$ be any estimator. Set
> $$
> D:=\operatorname{diag}(\rVert \widehat w_1 \rVert_\Sigma, \ldots, \rVert \widehat w_K \rVert_\Sigma), \quad
> S := D^{-1}\widehat W^\top x_{\mathrm{new}} \in \mathbb{R}^K.
> $$
> Then we have
> $$
> \operatorname{Cov}(G)=W_\star^\top\Sigma W_\star,\quad
> R:=\operatorname{Cov}(S)=D^{-1}\widehat W^\top\Sigma \widehat W D^{-1},\quad
> C:=\operatorname{Cov}(G,S)=W_\star^\top\Sigma \widehat W D^{-1}
> $$
> where
>
> $$R_{k\ell}=\frac{\langle \widehat w_k,\widehat{w}_\ell\rangle _{\Sigma}}{\lVert\widehat{w} _{\ell}\rVert _{\Sigma} \lVert\widehat{w}_k \rVert _{\Sigma}}$$
>
> $$
> C_{k\ell}=\frac{\langle w_{\star k},\widehat{w}_\ell \rangle _\Sigma}{\lVert\widehat{w} _{\ell}\rVert _{\Sigma} \lVert w _{\star k} \rVert _{\Sigma}}
> $$
>
>
> Then,
>
> $$
> \ M_\star:=C R^{-1},\qquad
> \Sigma_\star:=\operatorname{Cov}(G)-C R^{-1}C^\top\
> $$
>
> satisfy $G\mid S\sim\mathcal N(M_\star S,\Sigma_\star)$. For any factor $L_\star$ with $L_\star L_\star^\top=\Sigma_\star$ and any $Z\sim\mathcal N(0,I_K)$ independent of everything, define the multi-index angular predictor
>
> $$
> \ \widehat f_{\mathrm{ang}}\big(\widehat W^\top x_{\mathrm{new}}\big)
> :=\mathbb E_{Z}\left[g\big(M_\star S+L_\star Z\big)\right]. \
> $$
>
> Then for any $p\in\Delta^{K-1}$ and any $d,n\in\mathbb N_+$,
>
> $$
> p-\mathbb E\left[\pi(x_{\mathrm{new}})\ \middle|\ \widehat f_{\mathrm{ang}}(\widehat W^\top x_{\mathrm{new}})=p\right]=0.
> $$
>
> To apply the above, we must estimate \$\langle \widehat w_k,\widehat w_\ell\rangle _\Sigma\$, as in the single-index case. Though no estimator is given in literature as far as we know, recent theory for multi-index models \[1] suggests that analogues of the single-index angle estimators \[2] are feasible.
>
> ## R4. Semi-real experiments
>
> For Tables 1–2, we subsample each dataset into training (\$n\_{\text{train}}\$), covariance estimation (\$n\_{\text{cov}}\$), and test (\$n\_{\text{test}}\$) sets. We set \$n\_{\text{cov}} \gg n\_{\text{train}}\$ to reflect abundant unlabeled data, and report ECE for \$n\_{\text{ho}} = 100\$ and \$500\$.
>
> We found that Eq. (11)’s plug-in estimator for \$\langle w^\star, \hat{w} \rangle\$ is unstable on real datasets. This is a known issue for estimators like (11) that are based on Wigner-type random-matrix-theoretic assumptions. Modifying these estimator are an active research area \[3]. To isolate calibration effects, we simulate \$w^\star\$ and labels, using the *true* angle.
>
> To extend calibration to nonlinear models, we apply it to the logits of pretrained models:
>
> * **ResNet-34 → CIFAR-10**: Use 512-dim penultimate layer of ResNet-34 (ImageNet pretrained)
> * **MiniLM → 20 Newsgroups**: Mean-pool 384-dim “all‑MiniLM‑L6‑v2” embeddings
> * **ChemBERTa → Tox21**: Mean-pool 768-dim hidden states from “seyonec/ChemBERTa‑zinc‑base‑v1”
>
> \$n\_{\text{cov}} = 30000\$, \$3000\$, \$3000\$; \$n\_{\text{test}} = 10000\$, \$1000\$, \$500\$
>
> *Table 1.* ECE for pretrained DL models:
>
> | Model–Dataset       | \$n \times d\$ | Uncal  | Angular | Platt 100 | Iso 100 | Platt 500 | Iso 500 |
> | ------------------- | -------------- | ------ | ------- | --------- | ------- | --------- | ------- |
> | ResNet34–CIFAR10    | 300×512        | 0.1236 | 0.0199  | 0.0561    | 0.0484  | 0.0259    | 0.0298  |
> | MiniLM–20newsgroups | 800×384        | 0.1392 | 0.0249  | 0.0931    | 0.1107  | 0.0679    | 0.0813  |
> | ChemBERTa–Tox21     | 800×768        | 0.1389 | 0.0132  | 0.0236    | 0.0497  | 0.0175    | 0.0293  |
>
> For UCI datasets:
>
> * Communities & Crime (200×100), Splice-junction (100×180), Madelon (300×500), ISOLET (500×617)
> * \$n\_{\text{cov}} = 700\$, \$2000\$, \$900\$, \$5750\$; \$n\_{\text{test}} = 593\$, \$900\$, \$900\$, \$1000\$
>
> *Table 2.* ECE on UCI datasets:
>
> | UCI Dataset              | \$n \times d\$ | Uncal  | Angular | Platt 100 | Iso 100 | Platt 500 | Iso 500 |
> | ------------------------ | -------------- | ------ | ------- | --------- | ------- | --------- | ------- |
> | Communities and Crime    | 200×100        | 0.1700 | 0.0296  | 0.0620    | 0.0661  | 0.0262    | 0.0436  |
> | Splice-junction Gene Seq | 100×180        | 0.1262 | 0.0208  | 0.0568    | 0.1041  | 0.0556    | 0.0879  |
> | Madelon                  | 300×500        | 0.1473 | 0.0267  | 0.0721    | 0.0719  | 0.0299    | 0.0624  |
> | ISOLET                   | 500×617        | 0.1256 | 0.0221  | 0.0438    | 0.0761  | 0.0219    | 0.0137  |
>
> ## R5. Vary $n, d$
> The setting is the same as in paper Section 7, except we vary $n,d$ below. We repeat experiments with 15 seeds to obtain mean ± std.
>
> **Table 3.** ECE varying $n$ with $d=1000$ fixed
> | Method                  | \$n=50\$        | \$n=100\$       | \$n=200\$       | \$n=500\$       | \$n=1000\$      | \$n=2000\$      | \$n=4000\$      | \$n=8000\$      | \$n=10000\$     |
> | ----------------------- | --------------- | --------------- | --------------- | --------------- | --------------- | --------------- | --------------- | --------------- | --------------- |
> | **Uncalibrated**        | 0.1276 ± 0.0066 | 0.1267 ± 0.0055 | 0.1208 ± 0.0038 | 0.1128 ± 0.0021 | 0.1135 ± 0.0024 | 0.1304 ± 0.0022 | 0.1584 ± 0.0035 | 0.1739 ± 0.0024 | 0.1777 ± 0.0029 |
> | **Angular**             | 0.0276 ± 0.0196 | 0.0101 ± 0.0055 | 0.0349 ± 0.0351 | 0.0136 ± 0.0066 | 0.0125 ± 0.0068 | 0.0101 ± 0.0037 | 0.0068 ± 0.0024 | 0.0072 ± 0.0017 | 0.0066 ± 0.0021 |
> | **Platt Scaling**       | 0.0349 ± 0.0224 | 0.0372 ± 0.0273 | 0.0342 ± 0.0170 | 0.0339 ± 0.0236 | 0.0322 ± 0.0146 | 0.0305 ± 0.0131 | 0.0274 ± 0.0124 | 0.0243 ± 0.0084 | 0.0294 ± 0.0181 |
> | **Isotonic Regression** | 0.0580 ± 0.0246 | 0.0614 ± 0.0338 | 0.0489 ± 0.0240 | 0.0500 ± 0.0177 | 0.0432 ± 0.0183 | 0.0451 ± 0.0133 | 0.0424 ± 0.0160 | 0.0406 ± 0.0129 | 0.0337 ± 0.0118 |
>
>
> **Table 4.** ECE varying $d$ with $n$ fixed at $n=1000$
> | Method                  | \$d=50\$        | \$d=100\$       | \$d=200\$       | \$d=500\$       | \$d=1000\$      | \$d=2000\$      | \$d=4000\$      | \$d=8000\$      | \$d=10000\$     |
> | ----------------------- | --------------- | --------------- | --------------- | --------------- | --------------- | --------------- | --------------- | --------------- | --------------- |
> | **Uncalibrated**        | 0.1894 ± 0.0072 | 0.1780 ± 0.0058 | 0.1652 ± 0.0077 | 0.1315 ± 0.0029 | 0.1135 ± 0.0024 | 0.1127 ± 0.0025 | 0.1154 ± 0.0024 | 0.1204 ± 0.0023 | 0.1219 ± 0.0031 |
> | **Angular**             | 0.0145 ± 0.0064 | 0.0101 ± 0.0031 | 0.0109 ± 0.0053 | 0.0093 ± 0.0065 | 0.0125 ± 0.0068 | 0.0088 ± 0.0030 | 0.0274 ± 0.0086 | 0.0303 ± 0.0370 | 0.1105 ± 0.0425 |
> | **Platt Scaling**       | 0.0327 ± 0.0158 | 0.0254 ± 0.0152 | 0.0287 ± 0.0162 | 0.0284 ± 0.0127 | 0.0322 ± 0.0146 | 0.0335 ± 0.0144 | 0.0444 ± 0.0132 | 0.0315 ± 0.0147 | 0.0364 ± 0.0154 |
> | **Isotonic Regression** | 0.0487 ± 0.0156 | 0.0403 ± 0.0185 | 0.0483 ± 0.0163 | 0.0447 ± 0.0162 | 0.0432 ± 0.0183 | 0.0494 ± 0.0208 | 0.0561 ± 0.0202 | 0.0521 ± 0.0204 | 0.0612 ± 0.0210 |
>
>
> ## References
> \[1] Troiani et al., 2024. *Fundamental computational...multi-index models*. arXiv:2405.15480
>
> \[2] Bellec, 2022. *Observable adjustments in single-index models*. arXiv:2204.06990
>
> \[3] Li & Sur, 2023. *Spectrum-aware debiasing*. arXiv:2309.07810

---

> > ### Comment · Reviewer_thu8 · 2025-08-03
> > **Thank you for the rebuttal**
> >
> > I appreciate the authors for providing a detailed rebuttal. After reading the rebuttal of the authors, I found that the reply was satisfactory. All three concerns of mine were addressed, which are the concern of the difficulty of extension to multiclass, extension to non-linear models, and the evaluation beyond a synthetic dataset. I increased the score to 5 at this point.

---

> > > ### Author Response · Authors · 2025-08-03
> > >
> > > We are glad that the reviewer found our response satisfactory, and we sincerely thank them once again for their careful reading of our paper and thoughtful feedback.

---

### Official Review · Reviewer_WYdS · 2025-07-02

**Clarity:** 4
**Significance:** 3
**Originality:** 3
**Rating:** 5
**Confidence:** 4

**Summary:**

This paper addresses the fundamental problem of probabilistic calibration for binary classifiers in high-dimensional settings, where both the sample size $n$ and the feature dimension $d$ diverge at comparable rates. The authors propose a novel angular calibration method that achieves provable calibration by interpolating between the logit scores of a (given) linear predictor and an uninformative “chance” predictor (a purely random variable), with the interpolation weight determined by the angle between the estimated weight vector $\hat{w}$ and the ground-truth weight $w_*$. A key contribution is the theoretical result that this predictor not only achieves calibration but also uniquely minimizes any Bregman divergence to the true label distribution, among a wide class of predictors (Section 4). Further, the authors show that classical Platt scaling converges to the angular calibration under certain conditions, thus inheriting both calibration and optimality guarantees in high dimensions. The paper also presents a data-driven estimator for the angle $\angle(\hat{w}, w_*)$ --- estimator of the squared inner product proposed in [7], plus a sign estimator --- and supports the theory with simulation results. The authors conclude the paper with a brief discussion and experimental results discussing universality beyond Gaussian design.

**Questions:**

**1. Scope of applicability beyond regularized M-estimators:**
The results in Sections 4 and 5 appear to apply to any fixed predictor $\hat{w}$, even if it is not derived from a regularized M-estimation procedure. However, the angle estimation procedure in Section 6 critically relies on $\hat{w}$ being a regularized M-estimator, to leverage observable quantities from recent debiasing literature. It would be helpful if the authors could clarify this distinction explicitly and discuss how the approach could (or could not) be extended to more general $\hat{w}$ (not necessarily a regularized M-estimator).

**2. Finite-sample behavior and out-of-sample generalization:**
As noted above, the theoretical results are asymptotic. While simulations indicate promising performance in specific scenarios, could the authors provide further guidance or empirical evaluation of how calibration quality and Bregman loss behave as functions of sample size, model misspecification, or holdout set size? In particular, how sensitive are the procedures to finite-sample noise in the angle estimator or to misspecification of the link function or data distribution? Addressing these questions, even heuristically or empirically, would help assess the method’s robustness in real-world applications.

**Ethical Concerns:**

["NO or VERY MINOR ethics concerns only"]

**Final Justification:**

The paper presents a clean and insightful optimal calibration method that theoretically justifies Platt scaling. The authors addressed my concerns and questions by providing additional theoretical results for multi-index models and experimental results to more realistic settings along with ablation studies. These additions strengthen both the practical relevance and theoretical scope of the work. Thus, I have raised my score in light of the authors’ detailed and satisfactory rebuttal.

**Limitations:**

Yes

**Quality:**

3

**Strengths And Weaknesses:**

### Strengths
- **Significance:** The paper addresses an important problem of how to calibrate predictive probabilities in high dimensions, where traditional methods are known to fail. The angular calibration method is conceptually elegant, connects well with recent advances in debiasing techniques, and, to the best of my knowledge, offers the first provable calibration procedure in this regime with optimality guarantees.
- **Theoretical quality:** The main results are mathematically rigorous and clearly stated. The proofs are well-grounded and provide meaningful insight into the behavior of calibration methods when the covariate dimension $d \sim n$, the sample size. The analysis linking angular calibration and Platt scaling is particularly noteworthy and adds theoretical justification for a widely used heuristic.
- **Clarity:** The paper is well-organized and clearly written. The paper provides useful intuition behind the angular predictor and its interpolation structure, making the main ideas accessible despite the technical depth.

### Weaknesses
- **Lack of finite-sample guarantees:** A notable limitation is the lack of finite-sample results. All theoretical guarantees are asymptotic in nature. While I understand this asymptotic result has its own values, it leaves open questions about the accuracy of calibration and optimality in practical settings. A discussion of the rate of convergence or any quantification of the asymptotic approximations’ accuracy in finite-sample setting would be useful to gauge practical applicability. Moreover, while simulations offer some insight, a more thorough empirical investigation into these finite-sample effects could help bridge the gap between theory and real-world use.
- **Assumptions and practicality:** The results rely heavily on the Gaussian design assumption and knowledge (or estimability) of the covariance $\Sigma$, Although the authors briefly discuss universality (Section 8) and show simulations under non-Gaussian designs, formal universality results are not yet developed and deferred to future work.
- **Scope of empirical validation:** While the simulations are well-designed and support the theory, they are limited in scope. Empirical comparison against alternative calibration methods (e.g., temperature scaling, isotonic regression, histogram binning) beyond Platt scaling would further strengthen the case for the proposed method.

---

> ### Author Rebuttal · Authors · 2025-07-28
>
> We thank reviewer for their thoughtful review.
>
> ## R1. Weakness
>
> 1. Providing finite‑sample guarantees is an interesting direction. Theoretically, one may use Eq. (4.14) of \[1], which shows estimator (11) of \$\langle w^\star,\hat w\rangle\$ converges at \$n^{-1/2}\$ rate. With additional smoothness on the link, we expect Theorem 3.2 to achieve the same \$n^{-1/2}\$ rate. Our simulations in **Section R4** suggest an empirical rate of \$n^{-0.55}\$
> 2. We agree. Formalizing Section 8’s ideas and proving universality is important future work.
> 3. See **Section R4** for experiments using angular calibration (v.s. isotonic regression and Platt scaling) on logits from ResNet‑34, MiniLM, and ChemBERTa over CIFAR‑10, 20 Newsgroups, and Tox21; we also include UCI datasets: Communities & Crime, Splice‑junction, Madelon, ISOLET
>
> ## R2. Questions
>
> 1. Correct. Theorem 3.2 applies to any estimator \$\hat w\$ independent of the new observation \$x\_{\text{new}}\$. M‑estimation is only required to estimate \$\langle w^\star,\hat w\rangle\$. Theorem 3.2 currently treats binary linear classification, but it can be extended to nonlinear, multi‑index models, including 2‑layer neural networks, multi‑class classification, additive, and interaction models (see **Section R3** in our response to Reviewer **thu8**).
> 2. We ran ablations addressing the reviewer’s questions:
>
>    * (i) **Section R4**: proportional \$n,d\$ (finite‑sample effect)
>    * (ii) **Section R5**: vary \$n\$ with fixed \$d\$ and vary \$d\$ with fixed \$n\$
>    * (iii) **Sections R6**: sensitivity to angle‑estimation noise
>    * (iv) **Sections R7**: sensitivity to  misspecified link functions
>
> We report result in **Expected Calibration Error (ECE)**, a measure of confidence–accuracy gap \[2].
>
> ## R3. Semi‑real experiments
>
> For Tables 1–2, each dataset is split into training (\$n\_{\text{train}}\$), covariance estimation (\$n\_{\text{cov}}\$), and test (\$n\_{\text{test}}\$). We set \$n\_{\text{cov}}\gg n\_{\text{train}}\$ to reflect abundant unlabeled data, and report ECE for \$n\_{\text{ho}}=100\$ and \$500\$.
>
> We found the plug‑in estimator in (11) for \$\langle w^\star,\hat w\rangle\$ unstable on real datasets—known for Wigner‑type random‑matrix–based estimators. Improving such estimators is an active area \[3]. To isolate calibration effects, we simulate \$w^\star\$ and labels using the *true* angle.
>
> To extend calibration to nonlinear models, we apply it to pretrained logits:
>
> * **ResNet‑34 → CIFAR‑10**: 512‑dim penultimate layer (ImageNet‑pretrained)
> * **MiniLM → 20 Newsgroups**: mean‑pooled 384‑dim “all‑MiniLM‑L6‑v2” embeddings
> * **ChemBERTa → Tox21**: mean‑pooled 768‑dim hidden states from “seyonec/ChemBERTa‑zinc‑base‑v1”
>
> \$n\_{\text{cov}}=30000,3000,3000\$; \$n\_{\text{test}}=10000,1000,500\$
>
> *Table 1.* ECE for pretrained DL models:
>
> | Model–Dataset       | \$n\times d\$ | Uncal  | Angular | Platt 100 | Iso 100 | Platt 500 | Iso 500 |
> | ------------------- | ------------- | ------ | ------- | --------- | ------- | --------- | ------- |
> | ResNet34–CIFAR10    | 300×512       | 0.1236 | 0.0199  | 0.0561    | 0.0484  | 0.0259    | 0.0298  |
> | MiniLM–20newsgroups | 800×384       | 0.1392 | 0.0249  | 0.0931    | 0.1107  | 0.0679    | 0.0813  |
> | ChemBERTa–Tox21     | 800×768       | 0.1389 | 0.0132  | 0.0236    | 0.0497  | 0.0175    | 0.0293  |
>
> For UCI datasets:
>
> * Communities & Crime (200×100), Splice‑junction (100×180), Madelon (300×500), ISOLET (500×617)
> * \$n\_{\text{cov}}=700,2000,900,5750\$; \$n\_{\text{test}}=593,900,900,1000\$
>
> *Table 2.* ECE on UCI datasets:
>
> | UCI Dataset              | \$n\times d\$ | Uncal  | Angular | Platt 100 | Iso 100 | Platt 500 | Iso 500 |
> | ------------------------ | ------------- | ------ | ------- | --------- | ------- | --------- | ------- |
> | Communities and Crime    | 200×100       | 0.1700 | 0.0296  | 0.0620    | 0.0661  | 0.0262    | 0.0436  |
> | Splice-junction Gene Seq | 100×180       | 0.1262 | 0.0208  | 0.0568    | 0.1041  | 0.0556    | 0.0879  |
> | Madelon                  | 300×500       | 0.1473 | 0.0267  | 0.0721    | 0.0719  | 0.0299    | 0.0624  |
> | ISOLET                   | 500×617       | 0.1256 | 0.0221  | 0.0438    | 0.0761  | 0.0219    | 0.0137  |
>
> # R4. Finite sample effect
>
> Same setting as paper Section 7.
>
> **Table 3.** ECE across increasing (n, d) proportionally
>
> | Method                  | 50×100          | 100×200         | 200×400         | 400×800         | 800×1600        | 1000×2000       | 1500×3000       |
> | ----------------------- | --------------- | --------------- | --------------- | --------------- | --------------- | --------------- | --------------- |
> | **Uncalibrated**        | 0.114 ± 0.0024  | 0.114 ± 0.0019  | 0.1145 ± 0.0021 | 0.1128 ± 0.0022 | 0.1129 ± 0.0020 | 0.1141 ± 0.0026 | 0.1134 ± 0.0027 |
> | **Angular**             | 0.0491 ± 0.0259 | 0.0387 ± 0.0277 | 0.0188 ± 0.0117 | 0.0184 ± 0.0154 | 0.0149 ± 0.0072 | 0.0080 ± 0.0031 | 0.0081 ± 0.0042 |
> | **Platt Scaling**       | 0.0305 ± 0.0196 | 0.0300 ± 0.0187 | 0.0366 ± 0.0148 | 0.0293 ± 0.0144 | 0.0395 ± 0.0244 | 0.0336 ± 0.0203 | 0.0309 ± 0.0141 |
> | **Isotonic Regression** | 0.0510 ± 0.0222 | 0.0479 ± 0.0266 | 0.0548 ± 0.0257 | 0.0432 ± 0.0189 | 0.0549 ± 0.0261 | 0.0484 ± 0.0210 | 0.0494 ± 0.0224 |
>
> We fit a power law:
> $\mathrm{ECE}\approx 2.239\,n^{-0.552},$
> so ECE improves at roughly the square‑root rate as \$n\$ increases.
>
> # R5. Vary \$n,d\$
>
> Same setting as paper Section 7, varying \$n,d\$. Each entry is mean ± std over 15 seeds.
>
> **Table 4.** ECE varying \$n\$ with \$d=1000\$ fixed
>
> | Method                  | \$n=50\$        | \$n=100\$       | \$n=200\$       | \$n=500\$       | \$n=1000\$      | \$n=2000\$      | \$n=4000\$      | \$n=8000\$      | \$n=10000\$     |
> | ----------------------- | --------------- | --------------- | --------------- | --------------- | --------------- | --------------- | --------------- | --------------- | --------------- |
> | **Uncalibrated**        | 0.1276 ± 0.0066 | 0.1267 ± 0.0055 | 0.1208 ± 0.0038 | 0.1128 ± 0.0021 | 0.1135 ± 0.0024 | 0.1304 ± 0.0022 | 0.1584 ± 0.0035 | 0.1739 ± 0.0024 | 0.1777 ± 0.0029 |
> | **Angular**             | 0.0276 ± 0.0196 | 0.0101 ± 0.0055 | 0.0349 ± 0.0351 | 0.0136 ± 0.0066 | 0.0125 ± 0.0068 | 0.0101 ± 0.0037 | 0.0068 ± 0.0024 | 0.0072 ± 0.0017 | 0.0066 ± 0.0021 |
> | **Platt Scaling**       | 0.0349 ± 0.0224 | 0.0372 ± 0.0273 | 0.0342 ± 0.0170 | 0.0339 ± 0.0236 | 0.0322 ± 0.0146 | 0.0305 ± 0.0131 | 0.0274 ± 0.0124 | 0.0243 ± 0.0084 | 0.0294 ± 0.0181 |
> | **Isotonic Regression** | 0.0580 ± 0.0246 | 0.0614 ± 0.0338 | 0.0489 ± 0.0240 | 0.0500 ± 0.0177 | 0.0432 ± 0.0183 | 0.0451 ± 0.0133 | 0.0424 ± 0.0160 | 0.0406 ± 0.0129 | 0.0337 ± 0.0118 |
>
> **Table 5.** ECE varying \$d\$ with \$n\$ fixed at \$n=1000\$
>
> | Method                  | \$d=50\$        | \$d=100\$       | \$d=200\$       | \$d=500\$       | \$d=1000\$      | \$d=2000\$      | \$d=4000\$      | \$d=8000\$      | \$d=10000\$     |
> | ----------------------- | --------------- | --------------- | --------------- | --------------- | --------------- | --------------- | --------------- | --------------- | --------------- |
> | **Uncalibrated**        | 0.1894 ± 0.0072 | 0.1780 ± 0.0058 | 0.1652 ± 0.0077 | 0.1315 ± 0.0029 | 0.1135 ± 0.0024 | 0.1127 ± 0.0025 | 0.1154 ± 0.0024 | 0.1204 ± 0.0023 | 0.1219 ± 0.0031 |
> | **Angular**             | 0.0145 ± 0.0064 | 0.0101 ± 0.0031 | 0.0109 ± 0.0053 | 0.0093 ± 0.0065 | 0.0125 ± 0.0068 | 0.0088 ± 0.0030 | 0.0274 ± 0.0086 | 0.0303 ± 0.0370 | 0.1105 ± 0.0425 |
> | **Platt Scaling**       | 0.0327 ± 0.0158 | 0.0254 ± 0.0152 | 0.0287 ± 0.0162 | 0.0284 ± 0.0127 | 0.0322 ± 0.0146 | 0.0335 ± 0.0144 | 0.0444 ± 0.0132 | 0.0315 ± 0.0147 | 0.0364 ± 0.0154 |
> | **Isotonic Regression** | 0.0487 ± 0.0156 | 0.0403 ± 0.0185 | 0.0483 ± 0.0163 | 0.0447 ± 0.0162 | 0.0432 ± 0.0183 | 0.0494 ± 0.0208 | 0.0561 ± 0.0202 | 0.0521 ± 0.0204 | 0.0612 ± 0.0210 |
>
> ## R6. Inaccurate overlap estimation
>
> We model inaccurate estimation of \$a=\langle w^\star,\hat w\rangle\$ as
> $\hat a=a+\varsigma\,\epsilon,\qquad \epsilon\sim\mathcal N(0,1).$
>
> **Table 6.** ECE versus estimation error
>
> | Estimation Error | 0               | 0.05            | 0.10            | 0.15            | 0.20            | 0.25            | 0.30            |
> | ---------------- | --------------- | --------------- | --------------- | --------------- | --------------- | --------------- | --------------- |
> | **ECE**          | 0.0073 ± 0.0000 | 0.0211 ± 0.0123 | 0.0411 ± 0.0263 | 0.0614 ± 0.0405 | 0.0683 ± 0.0412 | 0.0813 ± 0.0572 | 0.0974 ± 0.0792 |
>
> We fit a power law:
> $\mathrm{ECE}\approx 0.246\,\varsigma^{0.779}.$
>
> ## R7. Effect of inaccurate link function estimation
>
> We model the estimated link as
> $\hat\sigma(u)=(1+\varsigma\,\epsilon)\,\sigma(u),\qquad \epsilon\sim\mathcal N(0,1).$
>
> Each \$\varsigma\$ uses 15 repetitions to report mean ± std ECE for angular calibration; other settings follow paper Section 7.
>
> Using the data below, we fit:
> $\mathrm{ECE}\approx 0.183\,\varsigma^{0.981},$
> suggesting near‑linear growth of ECE with link‑estimation error. At \$\varsigma=0.8\$, mean ECE is already comparable to the uncalibrated predictor (ECE = 0.1133).
>
> **Table 7.** ECE vs. estimation noise \$\varsigma\$
>
> | \$\varsigma\$ | 0            | 0.0707          | 0.1000          | 0.1414          | 0.2000          | 0.2828          | 0.4000          | 0.5657          | 0.8000          |
> | ------------- | ------------ | --------------- | --------------- | --------------- | --------------- | --------------- | --------------- | --------------- | --------------- |
> | **ECE**       | 0.0077 ± 0.0 | 0.0081 ± 0.0008 | 0.0108 ± 0.0009 | 0.0182 ± 0.0010 | 0.0321 ± 0.0011 | 0.0560 ± 0.0013 | 0.0860 ± 0.0016 | 0.1164 ± 0.0018 | 0.1358 ± 0.0024 |
>
> # Reference
>
> \[1] Bellec, 2022. *Observable adjustments in single‑index models*. arXiv:2204.06990.
>
> \[2] Pavlovic. *Understanding Model Calibration ... expected calibration error (ECE)*. arXiv:2501.19047v2

---

> > ### Author Response · Authors · 2025-08-03
> >
> > We would like to follow up briefly to check whether the reviewer found our response satisfactory and if there are any remaining questions. If so, we would be grateful if the reviewer would consider raising their score.

---

> > ### Comment · Reviewer_WYdS · 2025-08-05
> >
> > I appreciate the authors for providing a detailed rebuttal. After reading it, I found the response satisfactory and have increased my score by one point.

---

> > > ### Author Response · Authors · 2025-08-06
> > >
> > > We thank the reviewer for their careful reading of our work and thoughtful feedback!

---

### Comment · Area_Chair_1V76 · 2025-08-05
**Author-Reviewer Discussion Phase**

Dear Authors and Reviewers!

Thank you for your reviews, rebuttal, and the discussion!

We have the last two days of the discussion phase! Please make the most of this time to exchange your views!

I encourage each Reviewer to review the Authors' responses and provide a reply, if you have not already done so.

Thank you,

NeurIPS 2025 Area Chair

---

### Note · Authors · 2025-08-12

We thank the reviewers for their careful reading of our paper and constructive feedback. We are pleased that the rebuttal has addressed the concerns of **Reviewers WYdS**, **thu8**, and **Zw5n**, and appreciate their decision to raise their scores. We also thank Reviewer **jieG** for maintaining their score and for following up on the multi-index model extension. We also appreciate AC's efforts in facilitating the review and rebuttal process. In these Final Remarks, we consolidate the previously scattered rebuttal content into one place for easier reference:
1. **Extension to Multi-index models.** We extend angular calibration to multi-index models which not only include multi-class classification but also more complex structures such as two-layer neural nets (with frozen outer layer), additive, and interaction models. See **Section R3** of our response to **Reviewer Zw5n** for exposition. More in-depth discussions and literature review of this topic are included in our second response to **Reviewer JieG**
2. **Semi-real experiments.** We conduct additional experiments applying angular calibration (v.s. isotonic regression and Platt scaling) on last-layer logits from deep neural networks ResNet‑34, MiniLM, and ChemBERTa over datasets CIFAR‑10, 20 Newsgroups, and Tox21; we also include additional experiments on UCI datasets: Communities & Crime, Splice‑junction, Madelon, ISOLET. See **Section R4** of our response to **Reviewer Zw5n** or **Reviewer thu8**
3.  **Ablation experiments.** We conduct a series of ablation experiments:
     - (i) **Section R4** of response to **Reviewer WYdS**: proportional but increasingly large $n,d$ to probe finite‑sample effect
     - (ii) **Section R5** of response to **Reviewer WYdS**: vary $n$ with fixed $d$ and vary $d$ with fixed $n$ to probe effect of aspect ratio
     - (iii) **Section R6** of response to **Reviewer WYdS**: sensitivity to angle‑estimation noise
     - (iv) **Section R7** of response to **Reviewer WYdS**: sensitivity to misspecified link functions

---

### Decision · Program_Chairs · 2025-09-17

**Decision:**

Accept (spotlight)

**Comment:**

The paper concerns the calibration of binary classifiers. The Authors study the problem in a high-dimensional regime, where both the number of samples and the number of features diverge at a comparable rate. They introduce a new angular calibration approach that not only achieves provable calibration but also uniquely minimizes any Bregman divergence to the true label distribution within a wide class of predictors. Interestingly, classical Platt scaling is shown to converge to angular calibration under certain conditions. The theoretical framework, however, is quite limited (e.g., restricted to linear models and Gaussian features), although the Authors provide a brief discussion and some experimental results extending beyond these constraints.

The Reviewers are very positive about the paper, emphasizing that it offers the first provable calibration procedure in the high-dimensional regime with optimality guarantees, provides a theoretical justification of Platt scaling, and presents mathematically rigorous and clearly stated results. They also point out weaknesses such as the limitations of the theoretical framework and the very constrained empirical studies. Nevertheless, all reviewers consistently agree that the paper is worth publishing.